# Reliable, scalable functional genetics in bloodstream-form *Trypanosoma congolense* in vitro and in vivo

Georgina Awuah-Mensah[1], Jennifer McDonald[1¤a], Pieter C. Steketee[2], Delphine Autheman[3], Sarah Whipple[1], Simon D'Archivio[1¤b], Cordelia Brandt[4], Simon Clare[4], Katherine Harcourt[4], Gavin J. Wright[3,5], Liam J. Morrison[2], Catarina Gadelha[1]*, Bill Wickstead[1]*

1 School of Life Sciences, University of Nottingham, Nottingham, United Kingdom, 2 The Roslin Institute, Royal (Dick) School of Veterinary Studies, University of Edinburgh, Edinburgh, United Kingdom, 3 Cell Surface Signalling Laboratory, Wellcome Sanger Institute, Cambridge, United Kingdom, 4 Pathogen Support Team, Wellcome Sanger Institute, Cambridge, United Kingdom, 5 Department of Biology, Hull York Medical School, York Biomedical Research Institute, University of York, York, United Kingdom

¤a Current address: Department of Pathology, University of Cambridge, United Kingdom
¤b Current address: Sygnature Discovery, Pennyfoot Street, Nottingham, United Kingdom
* catarina.gadelha@nottingham.ac.uk (CG); bill.wickstead@nottingham.ac.uk (BW)

**Data Availability Statement:** RNA-seq read data are available from the European Nucleotide Archive database (accession number ERP125374) link: https://www.ebi.ac.uk/ena/browser/view/

## Abstract

Animal African trypanosomiasis (AAT) is a severe, wasting disease of domestic livestock and diverse wildlife species. The disease in cattle kills millions of animals each year and inflicts a major economic cost on agriculture in sub-Saharan Africa. Cattle AAT is caused predominantly by the protozoan parasites *Trypanosoma congolense* and *T. vivax*, but laboratory research on the pathogenic stages of these organisms is severely inhibited by difficulties in making even minor genetic modifications. As a result, many of the important basic questions about the biology of these parasites cannot be addressed. Here we demonstrate that an in vitro culture of the *T. congolense* genomic reference strain can be modified directly in the bloodstream form reliably and at high efficiency. We describe a parental single marker line that expresses *T. congolense*-optimized T7 RNA polymerase and Tet repressor and show that minichromosome loci can be used as sites for stable, regulatable transgene expression with low background in non-induced cells. Using these tools, we describe organism-specific constructs for inducible RNA-interference (RNAi) and demonstrate knockdown of multiple essential and non-essential genes. We also show that a minichromosomal site can be exploited to create a stable bloodstream-form line that robustly provides >40,000 independent stable clones per transfection–enabling the production of high-complexity libraries of genome-scale. Finally, we show that modified forms of *T. congolense* are still infectious, create stable high-bioluminescence lines that can be used in models of AAT, and follow the course of infections in mice by in vivo imaging. These experiments establish a base set of tools to change *T. congolense* from a technically challenging organism to a routine model for functional genetics and allow us to begin to address some of the fundamental questions about the biology of this important parasite.

PRJEB41578. All other relevant data are within the manuscript and its Supporting Information files.

**Funding:** This work was supported by University of Nottingham/Wellcome Trust Institutional Strategic Support Fund 204843/Z/16/Z awards to BW and to CG; Sir Halley Stewart Medical Research Grant R410 to CG and GAM; BBSRC studentship 1364116 to JM; BBSRC grants BB/N007492/1, BB/S00243X/1, and BBS/E/D/20002173 to LJM and PCS; and Wellcome Trust grant 206194 and BBSRC grant BB/S001980/1 to DA, CB, SC, KH and GJW. The funders had no role in study design, data collection and analysis, decision to publish, or preparation of the manuscript.

**Competing interests:** The authors have declared that no competing interests exist.

## Author summary

The parasites *Trypanosoma congolense* and *T. vivax* are the most significant causative agents of Animal African trypanosomiasis (AAT). AAT kills an estimated 3 million cattle each year and represents a huge financial burden on food production in sub-Saharan Africa. A critical tool for understanding pathogen biology is the ability to make genetic modifications, especially creating specific mutants of target genes that can be used to investigate the locations of gene products, the effects of changes in expression, or consequence of complete gene removal. However, work on AAT is severely limited by difficulties in making even small genetic modifications and lack of tools for many functional genetics applications. Here, we design, test and validate a set of tools for *T. congolense* that brings for the first time: routine high-efficiency gene tagging and knockout, regulatable transgene expression from silent loci, a species-specific system for inducible gene knockdown, bioluminescent lines for in vivo disease models, and a means to generate highly complex libraries of mutants that will enable genome-scale work. These data and the tools around them will greatly aid research into AAT and *T. congolense* biology.

## Introduction

Animal African trypanosomiasis (AAT) is a parasitic disease associated with anaemia, loss of condition and death in sub-Saharan livestock. The impact of the disease on cattle farming is particularly severe. Each year, AAT causes ~3 million cattle deaths, with economic losses in cattle production alone estimated to be US$ 1–1.2 billion [1]. The disease in cattle is caused by trypanosomes of the species *Trypanosoma brucei* (sub-species of which also cause human disease), *T. vivax*, and *T. congolense*. Of these species, *T. congolense* may cause the majority of disease in sub-Saharan cattle [2,3], as well as making up a substantial proportion of infections [4].

In spite of the importance of *T. congolense* and *T. vivax* for AAT, the vast majority of laboratory research in African trypanosomes uses *T. brucei*, which is responsible for only a minor proportion of infections. However, there are known to be significant differences in the biology of these species, including in genomic content [5,6], antigenic variation [6–8], developmental progression [9], and disease symptoms and tropism [10,11], in addition to the differences in host specificity. There are also substantial differences in the differential regulation of genes involved in specific biological processes during infection [12] and between metabolomes (Steketee et al., in preparation). Such differences create a pressing need for species-specific models of AAT that can be used and modified in axenic culture and also transmitted through animals.

The focus of molecular research on *T. brucei* was historically driven by its association with human disease, but recent bias has been heavily influenced by the paucity of means for performing functional genetics in *T. congolense*. This is true for all lifecycle stages of *T. congolense*, but is particularly acute for bloodstream-form cells, where even generation of stable transformants has been difficult–necessitating laborious strategies for genetic modification involving multiple rounds of transfection and selection in procyclic cells followed by in vitro differentiation first to epimastigotes, then to metacyclic cells and finally to bloodstream forms (sometimes requiring passage through animals; [13,14]). Where transfections have been performed, they have been through application of plasmids designed and tested in *T. brucei* with little/no modification. Moreover, there has been no establishment of silent loci suitable for integration of inducible constructs, so although tetracycline-regulated gene expression has been demonstrated in *T. congolense* [14], the level of regulation was very poor (~4-fold), precluding their

use for applications such as expression of toxic gene products. Finally, work in *T. brucei* has been fundamentally changed by the development of methods that enable the generation of many thousands of transfectants from a single transfection [15], enabling the production of high-complexity libraries, such as in the RIT-Seq approach for high-throughput testing of RNA-interference mutants [16]. The lack of such methods in *T. congolense* severely restricts our ability to probe the mechanisms of pathogenicity, lifecycle control and resistance to drugs, amongst other critical questions.

Limitations in functional genetics tools for *T. congolense* not only restrict experimental work to species that may not be good models for AAT infections, but prevent the addressing of fundamental questions about the biology of these important species–including decoding the genetic basis for the differences seen in their pathology, virulence and resistance to drugs. In addition, since *T. congolense* is the only African trypanosome for which the whole lifecycle can be recapitulated in vitro by means of chemical cues alone [14], this organism has unique advantages as a model for studying differentiation across the African trypanosome clade. Here, taking as a start point a stable axenic bloodstream culture of the genome reference strain of *T. congolense* (IL3000), we have designed and implemented tools that allow direct, reliable transgenesis of multiple types in the pathogenic stage of this species. We demonstrate efficient modification of endogenous loci and gene knockout without the need for heterologous DNA modifiers, such as recombinases, restriction endonucleases or Cas9. We describe the design and use of a *T. congolense*-optimized construct to allow for inducible T7RNAP-driven expression and investigation of loci on the silent minichromosomes as sites for regulatable transgene expression. We also build and test new *T. congolense*-specific constructs for RNA-interference (RNAi) and describe a freeze/thaw-stable cell line enabling transfection efficiencies compatible with the production of genome-scale high-complexity libraries. Finally, we produce a stable, luminescent line that can be used in models of AAT in mice to increase efficiency of vaccinology and drug development studies.

## Results

### Routine high-efficiency transfection in bloodstream-form *T. congolense*

Although transfection of procyclic (insect midgut) forms of *T. congolense* is generally routine–albeit at low efficiency (~10 independent transfectants from a single transfection of ~$10^8$ procyclic cells; [17])–transfection of cultures of bloodstream-form *T. congolense* has been seen as challenging, with the majority of transfections giving no stable transfectants at all [14]. However, very few loci have been targeted to date, and published transfection attempts in bloodstream forms have used heterologous sequence from the *T. brucei* tubulin locus to direct integration (~400 bp homology at each end with ~10% mismatch to *T. congolense*), the impact of which on efficacy is unclear.

To test the efficiency of routine transfection directed by homologous targeting sequences in bloodstream-form *T. congolense*, we took the stable axenic bloodstream culture of the IL3000 strain of *T. congolense* originally derived by Coustou et al. ([14]; kindly provided by Michael Barrett, University of Glasgow) and targeted modifications to 9 different endogenous loci, either as modifications to the 5'- or 3'-end of the CDS, or as complete knockout of a gene. Unlike previous descriptions using heterologous sequence, 10 constructs with homologous sequence, covering 4 different plasmid architectures (see Materials and Methods), could be transfected at efficiencies more than sufficient for routine modification (100–4000 independent clones per transfection; Fig 1A). These efficiencies are at or slightly above those we routinely achieve targeting endogenous loci in *T. brucei* Lister 427 cells (typically ~1000 independent clones). The same preparations of DNA gave very similar numbers of clones

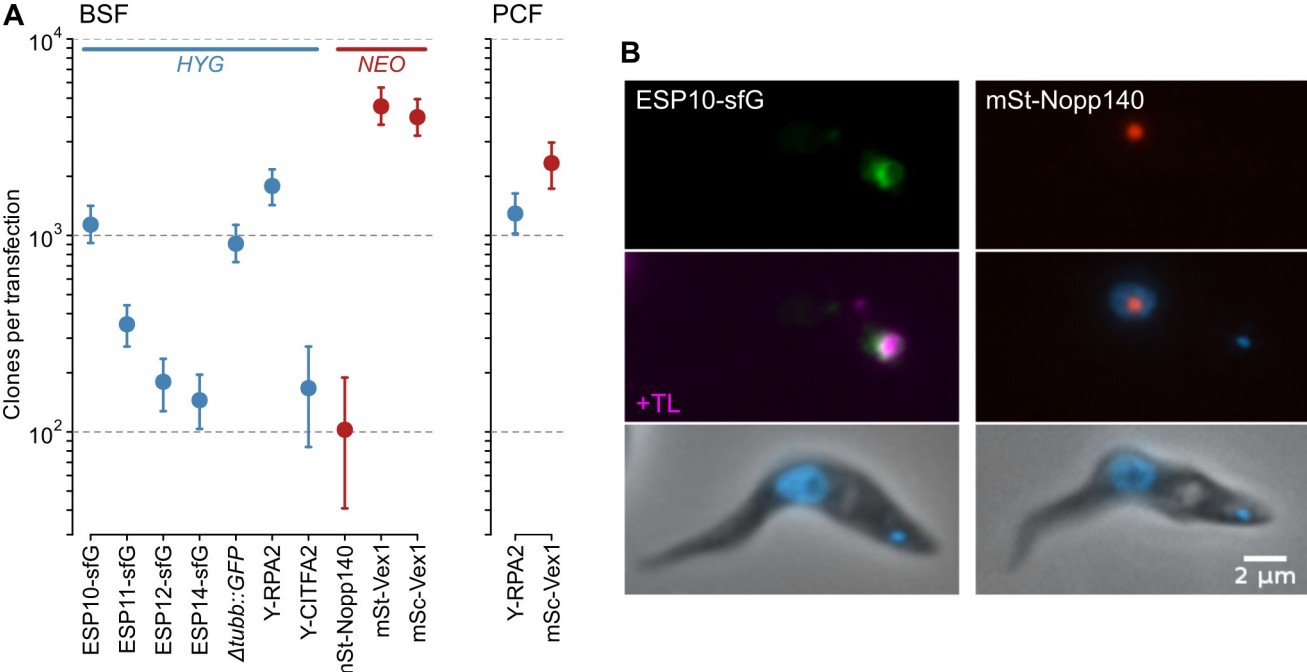

**Fig 1. High-efficiency modification of endogenous loci in bloodstream-form *T. congolense*.** A) Numbers of independent stable clones produced from electroporation of ~2.5×10⁷ cells with 10–15 µg linearised DNA targeting CDS encoding fluorescent proteins (sfG, superfolder GFP; Y, eYFP; mSt, mStrawberry; mSc, mScarlet-I) to endogenous loci in *T. congolense*. Modifications are either N-terminal tags (RPA2, CITFA2, Nopp140, Vex1), C-terminal tags (ESPs), or full replacement of a gene (*Δtubb::GFP*). Selection of integrants with genes conferring resistance to hygromycin (*HYG*) or G418 (*NEO*) give 100–5000 independent clones from a single transfection in both bloodstream-form (BSF) and procyclic-form (PCF) cells. Bars: 90% confidence intervals for estimates of clone numbers resulting from individual transfections (reflecting the total number of wells counted). B) Examples of localisation of expressed tagged proteins to expected cellular compartments in bloodstream-form *T. congolense*. *T. congolense* ESP10 homologue localises to the flagellar pocket region, visualised with fluorophore-conjugated tomato lectin (TL), while *T. congolense* Nopp140 localises to the nucleolus. Representative interphase cells are shown. See S1 Fig for larger fields of cells.

when transformed into procyclic-form IL3000 cells (Fig 1A). Moreover, in all clones tested (n = 2–8, depending on transfection), correct integration of DNA could be confirmed by detection of tagged protein or PCR, and sub-cellular localisations were in line with predictions based on putative function (see Fig 1B for examples). These data demonstrate that there is no intrinsic barrier to transgenesis in bloodstream-form *T. congolense* IL3000 and that specific loci can be selectively targeted with good efficiency.

## A 'single-marker' *T. congolense* line for regulated ectopic expression

Although regulatable systems based on IPTG [18], vanillic acid [19] and cumate [20] have been used in *T. brucei*, the tetracycline-based system is to date the best characterised and best regulated [21]. A bloodstream-form *T. congolense* cell line expressing bacterial tetracycline-repressor protein (TetR) and T7 bacteriophage RNA polymerase (T7RNAP) has been generated previously by *in vitro* differentiation of a procyclic IL3000 cells [14], but this line: i) uses 2 selection markers that would otherwise be available for further modification, ii) depends on heterologous sequences for processing of transgenes, and iii) does not include the codon-optimization used in the *T. brucei* SmOx lines [22] that has been shown to increase expression in this organism [23]. As a basis for regulated ectopic expression in *T. congolense*, we designed and built a single-marker construct (pTcoSM) that modifies the *T. congolense* tubulin locus by full replacement of a *TUBB* CDS with codon-optimized *T7RNAP* and *TetR*. Similarly to *T. brucei* pSmOx [22], pTcoSM integrates into the *T. congolense* genome without bacterial DNA to

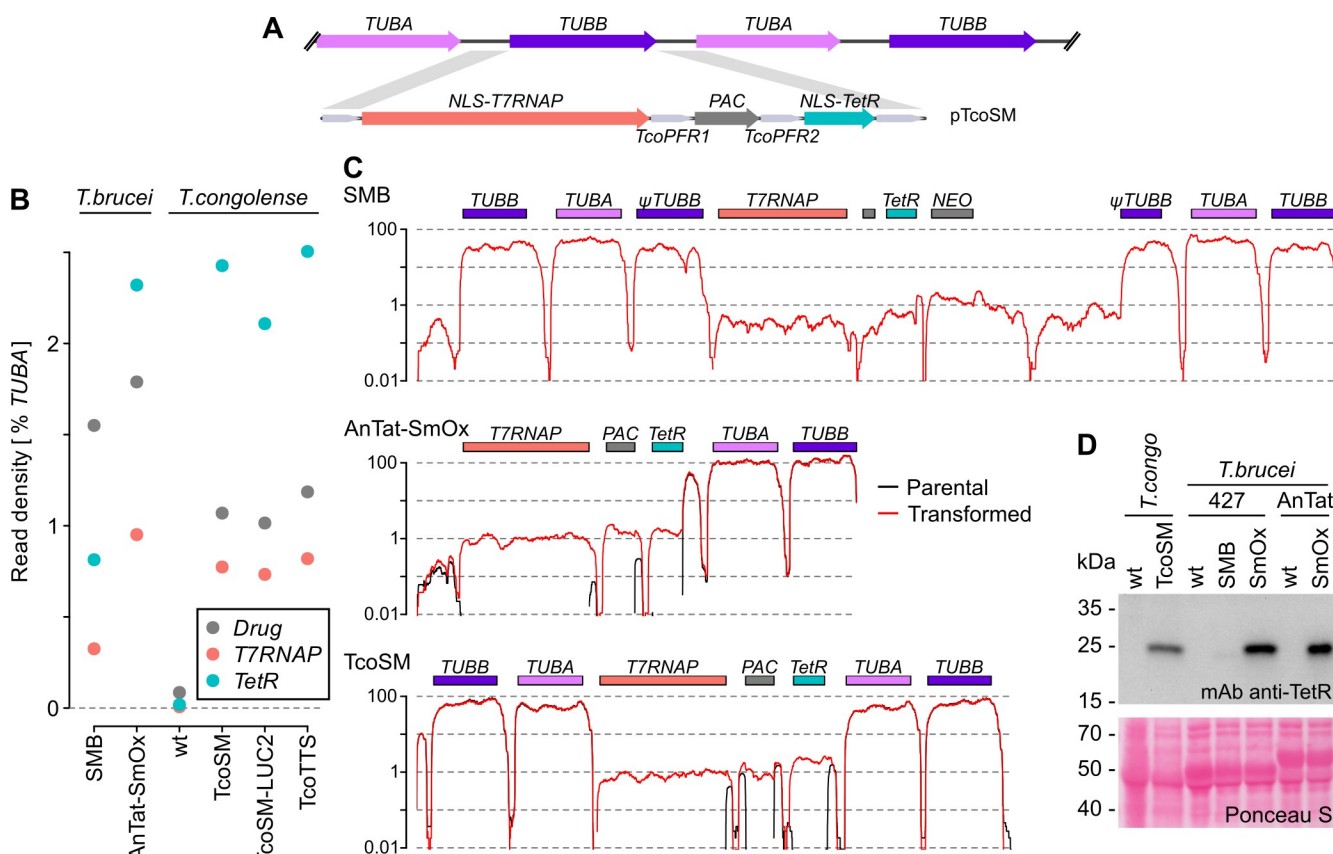

**Fig 2. Production of a 'single marker' bloodstream-form *T. congolense* line (TcoSM) for inducible expression of transgenes.** A) Schematic representation of genetic modification in TcoSM cells. *PAC* confers resistance to puromycin. *TcoPFR1* and *TcoPFR2* are the intergenic regions between copies of the native *T. congolense* IL3000 *PFR1* and *PFR2* arrays, respectively. B) Quantitation of mRNA levels for transgenes in TcoSM against common modifications of *T. brucei* (SMB, Wirtz et al. 'single-marker bloodstream' [21]; AnTat-SMOx, pleomorphic EATRO1125 bloodstream cells transformed with the Poon et al. pSmOx plasmid [22]; see Figs 8 and 10 for modifications producing TcoSM-LUC2 and TcoTTS cells). Read density is normalised to total reads resulting from *TUBA*. C) Read density across the modified regions in SMB, AnTat-SMOX and TcoSM cells. Peaks in intergenic regions in untransformed parental lines shows reads mapping to UTRs shared between integrated DNA and endogenous loci. D) Levels of transgenic TetR produced in modified cells assessed by immunoblotting of whole cell lysates with an anti-TetR monoclonal antibody. A section (encompassing the VSG region) of the same membrane stained with Ponceau S is shown as a control for loading. 'wt' indicates unmodified parental cells. The TetR protein in SMB cells lacks the N-terminal NLS used in AnTat-SMOx and TcoSM, so has a lower molecular mass. Data from one of two biological replicates are shown. See S2 Fig for full view of immunoblot membrane.

avoid potential mis-integration when using additional constructs containing common plasmid sequence, but in pTcoSM all introduced CDS are flanked by intergenic sequences associated with highly-expressed *T. congolense* loci (tubulin, PFR1 and PFR2; Fig 2A).

Transfection of pTcoSM into bloodstream-form IL3000 cells generated the *T. congolense* single marker line, TcoSM. Transcriptome analysis showed levels of *T7RNAP* and *TetR* mRNA in TcoSM are higher than in *T. brucei* 'single-marker bloodstream' cells [21], which have been very widely used in this organism, and similar to *T. brucei* lines carrying the pSmOx modification (Fig 2B and 2C), demonstrating both pol II read-through and processing of transgenes by the homologous intergenic sequences as expected. The discontinuation of commercial antibodies against T7RNAP precluded the direct testing of levels for this protein, but levels of TetR were also substantially above those in SMB cells (Fig 2D) which, together with the high mRNA levels, suggested levels of protein suitable for regulated expression.

## Inducible transgene expression from stable *T. congolense* minichromosomal loci

Essential for the development of well-regulated systems for inducible expression in *T. brucei* has been the identification of loci that are normally transcriptionally silent, but can support high-level transcription on induction. Lack of such identified loci is likely the major cause of the poor regulation (only 2.8- to 4.5-fold) seen in previous attempts at inducible transgene expression in *T. congolense*, as tetracycline-responsive elements were integrated into the active tubulin locus resulting in substantial expression in the absence of induction [14].

The most common integration target in *T. brucei* is a spacer region found between arrayed copies of the 18S/28S ribosomal RNA polycistron. Targeting this sequence in the opposite orientation to rRNA transcription can provide >1000-fold transgene regulation [21,24], although regulation appears to be highly dependent on the precise spacer at which integration occurs [24–26]. Unlike in *T. brucei*, the 18S/28S rRNA polycistrons in *T. congolense* do not appear to be arrayed [5], so there is no site homologous to the *T. brucei* rRNA spacer. Integration has been successfully directed to a non-homologous sequence upstream of the *T. congolense* rRNA promoter [27,28], but it is unclear if this site offers the same level of transcriptional regulation as the *T. brucei* rRNA spacer. As an alternative to sites on the housekeeping chromosomes, loci on the transcriptionally silent minichromosomes have been shown to provide sites for consistent, well-regulated expression in *T. brucei* [26,29]. As minichromosomes are even more abundant in *T. congolense* than in *T. brucei* [5] and transcriptomic data suggest them to be silent [12], we reasoned they might make good integration sites for regulated expression.

To test the utility of *T. congolense* minichromosomes as sites for regulated expression, we targeted a *GFP* and hygromycin resistance marker polycistron transcribed by a single tetracycline-responsive T7 promoter to 3 sequences present on minichromosomal contigs in the recent long-read assembly of the IL3000 genome [5]: i) the 369 bp repeat found abundantly on minichromosomes [30,31], ii) loci encoding an expanded family of putative DEAH-box RNA helicases (DBRH) related to Tb927.6.740 but present on minichromosomes in *T. congolense* (e.g. TcIL3000.A.H_000093500), and iii) a silent telomeric VSG gene (TcIL3000.A. H_000093600). 369 bp repeat and silent *VSG* were targeted with an efficiency indistinguishable from modifications made at pol II transcribed loci (Figs 1A and 3B). These efficiencies are consistent with integration into the *T. congolense* genome being independent of target copy number, as is also seen in *T. brucei* [32], as targets present in the long-read assemblies at 1, 30 and 13 000 copies per haploid genome direct integration with the same efficiency (Fig 3B). In contrast, transfection targeting *DBRH* produced ~20-fold fewer clones. This was independent of whether transgene transcription ran in the same or opposite orientation to the *DBRH* coding direction, showing that the effect was not a product of a specific DNA preparation, but likely an intrinsic property of either the targeting sequence (e.g. due to heterogeneity in sequence across the family) or inability of some of these loci to support transgene expression.

All successfully selected transfectants supported T7-driven expression at levels greatly above those from pol II read-through at the tubulin locus (Figs 3C and 4A). Mean protein levels were ~20-times higher when T7RNAP transcribed compared to pol II, which is at least as good as those from SMB or 90-13 *T. brucei* cells (~12 and 15-times, respectively; [21]), confirming that there is good production of T7RNAP in TcoSM cells. Removal of tetracycline reduced mean fluorescence of transfectants to background levels (>500-fold regulation at the protein level) for all except cells containing transgenes integrated into *DBRH* in the coding orientation, where there was substantial fluorescence in the absence of tetracycline (Figs 4A and S3). Interestingly, this does not appear to be due to direct basal transcription of *GFP* by

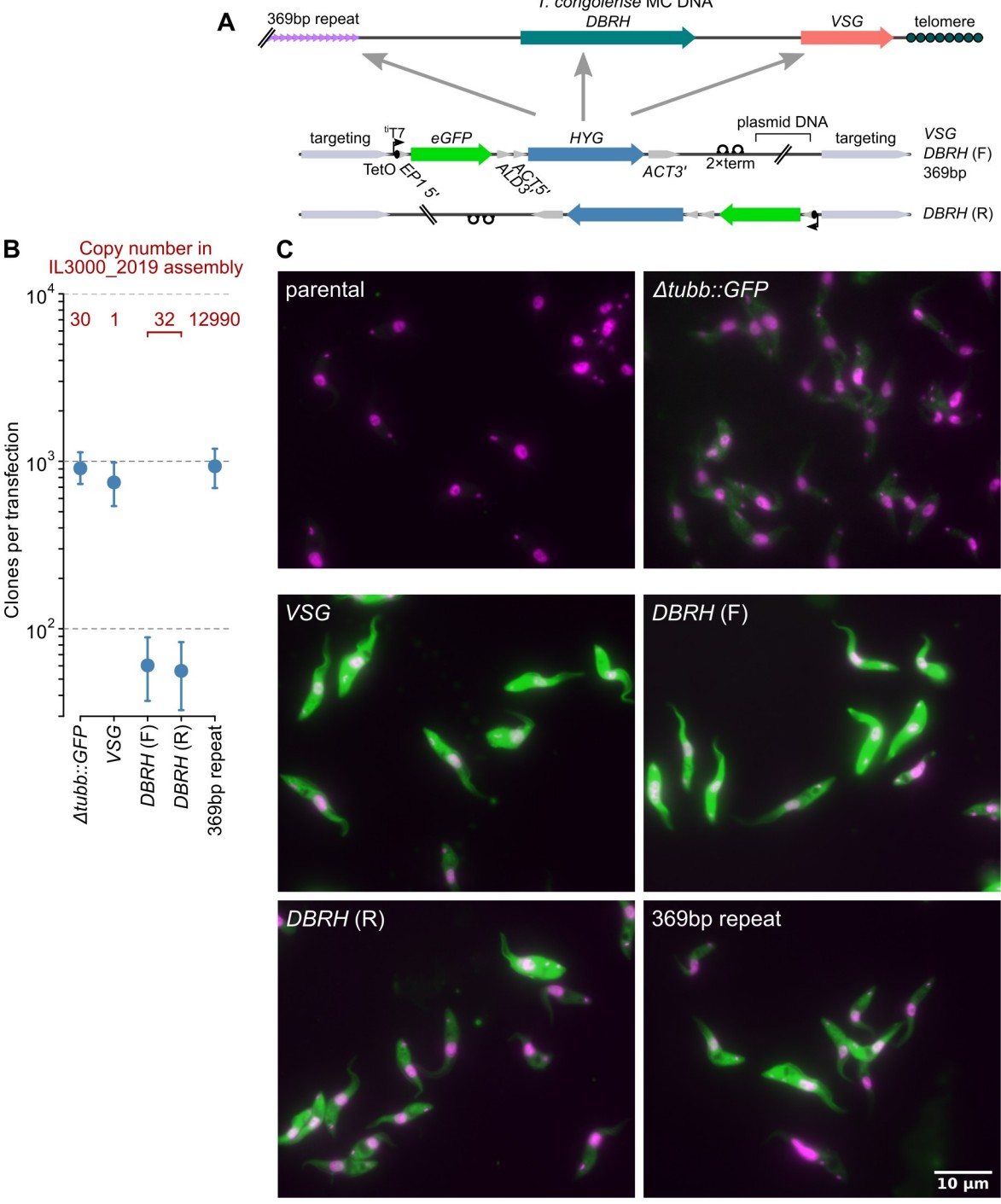

**Fig 3. Integration of inducible constructs at *T. congolense* minichromosomal loci.** A) Schematic representation of integrated construct and loci targeted. *VSG*, silent minichromosomal *VSG* TcIL3000.A.H_000093600; *DBRH*, gene encoding putative ATP-dependent DEAH-box RNA helicase (e.g. TcIL3000.A.H_000093500); 369bp repeat, minichromosome-specific satellite repeat [5,31]. Constructs targeting *DBRH* are identical other than integrating with inducible transcription running in the forward (F) or reverse (R) direction relative to *DBRH* CDS. B) Transfection efficiency when directing integration to minichromosomal or *TUBB* loci. 'Copy number' refers to number in the TriTrypDB *T. congolense* IL3000_2019 assemblies (v46) based on the work of [5]. Bars: 90% confidence intervals for estimates of clone numbers resulting from individual transfections. C) Expression of GFP (green) in cells in the presence of tetracycline. Counter-staining of cells with 4′,6-diamidino-2-phenylindole (DAPI; magenta) is also shown. All lines were captured and processed equally, except that GFP signal from *Δtubb*::*GFP* has been increased 6-fold for visualisation. Representative images from one clone for each modification are shown.

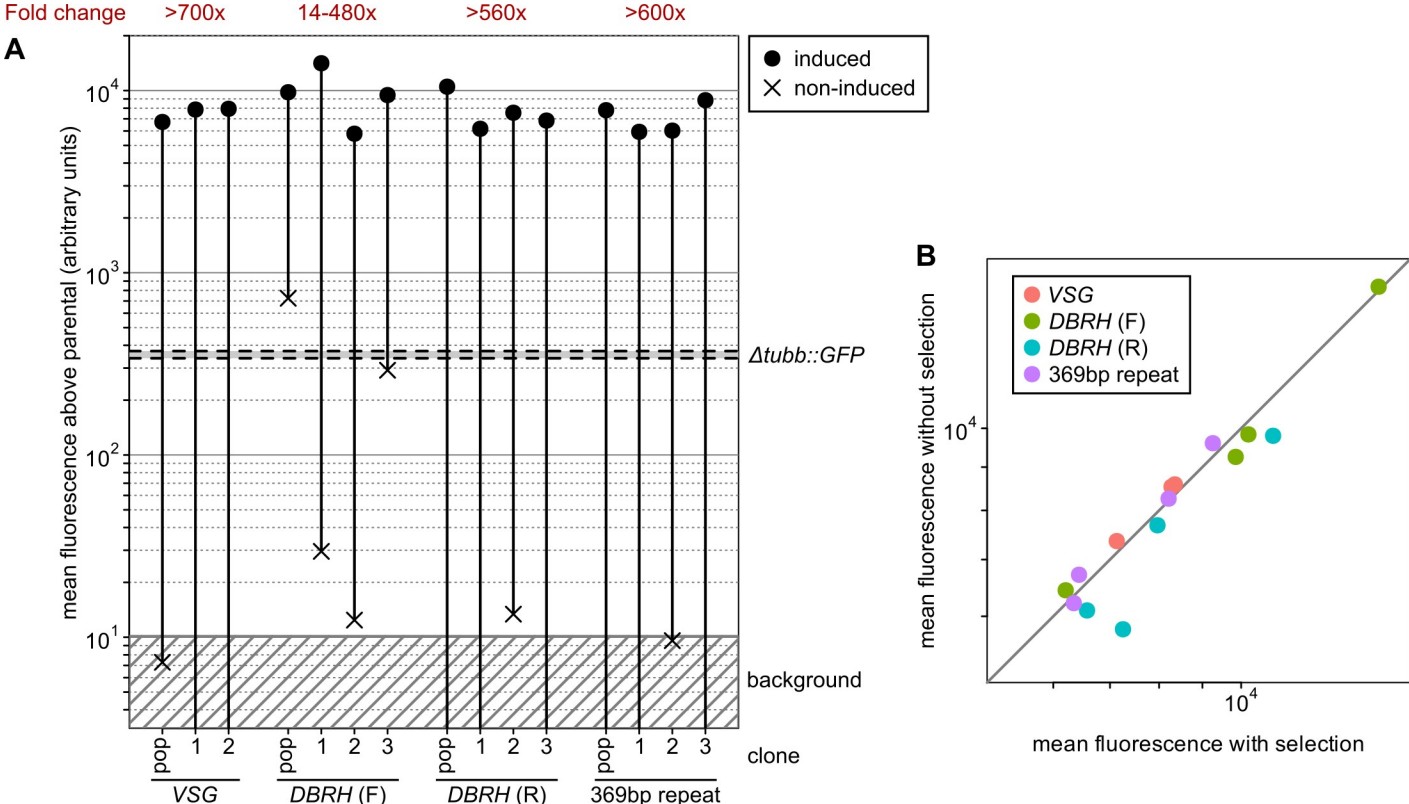

**Fig 4. Regulated expression from *T. congolense* minichromosomal loci.** Constructs and target sequences are as in Fig 3. A) Quantification of GFP regulation. Mean fluorescence in 0 (non-induced) or 1 µg ml⁻¹ (induced) tetracycline was calculated by flow cytometry and is shown as levels above mean background (untagged TcoSM cells). For comparison, range of triplicate measurements from 2 independent clones of *Δtubb::GFP* are also shown. Clone number is shown below x-axis. 'pop' indicates data from uncloned populations. Distributions of fluorescence for each cell line are shown in S3 Fig. B) Change in mean induced GFP levels following maintenance of cells for 21 days (~50 generations) without selection. Individual points represent same clones/populations as shown in Fig 3D and are from a single experimental replicate.

endogenous pol II, but to high-level GFP production in a sub-population of cells–presumably due to transient displacement of TetR (S3 Fig).

Induced GFP levels for 14 of 15 lines were essentially unchanged by growth for 21 days in the absence of transgene selection (Fig 4B), showing that most *T. congolense* minichromosomal loci have sufficient stability with respect to mitosis to also be good sites for transgenetic modifications where it may not be possible to constantly maintain selective drug (e.g. in vivo).

### Direct inducible RNA interference in *T. congolense* bloodstream forms

Due to difficulties in performing genetic modifications, no previous work has performed RNAi by direct modification of bloodstream-form *T. congolense*–instead making modifications (at actively transcribed loci) in procyclic cells and then differentiating to bloodstream forms, either with [13] or without [14] passage through animals. To take advantage of the ability to stably modify minichromosomal loci in bloodstream-form *T. congolense*, we designed 3 constructs for inducible RNAi that integrate at the silent minichromosomal *VSG*, TcIL3000.A. H_000093600. These constructs contain a tetracycline-responsive RNAi cassette slightly modified from p2T7$^{Ti}$ [33] that is widely used in *T. brucei*, a constitutively-expressed selection cassette derived from those providing the greatest transfection efficiency for endogenous-locus modification (Fig 1A), and targeting sequence to integrate at the silent *VSG* without bacterial

DNA (Fig 5A). Three different arrangements of promoter and terminator around the selection cassette were tested in the plasmids: p2T7-TcoV (*NEO*-expression driven by the *T. congolense* rRNA promoter), p3T7-TcoV (T7-driven *NEO* expression), and p3T7-TcoV$^{TT}$ (T7-driven expression terminated at end of cassette).

To test the efficiency of knockdown without the confounding influence of gene essentiality, we directed RNAi against a heterologous gene (*GFP*) integrated into the tubulin locus by inducible expression of ~600 bp *GFP* double-stranded RNA. Herein, this RNAi is indicated δ*gfp* by analogy to nomenclature used for gene knock-outs. All three *T. congolense*-specific RNAi plasmids reduced the abundance of GFP on induction (Figs 5B and S3), although knockdown was weaker in lines with pol I-driven selection than those with three T7 promoters (presumably as a result of transcriptional interference). There was no evidence at the protein level for leakiness of RNAi in the absence of tetracycline, but greatest knockdown was only to 30–40% of non-induced levels for the best performing construct (p3T7-TcoV). Similar levels of knockdown were seen when targeting an endogenous gene encoding flagellar pocket protein ESP14 [34]; although in *T. brucei*, RNAi against *ESP14* reduces proteins levels to an estimated <10% of parental, knockdown of the *T. congolense* orthologue (TcIL3000_7_180) is only to ~30% (Fig 5C and 5D). This effect is manifest at the RNA level, as RNAi targetted against a further 5 endogenous genes (*PPDK*, TcIL3000.A.H_000922100; *CHC*, TcIL3000.A.H_000768200; *FBPase*, TcIL3000.A.H_000671500; *FH1*, TcIL3000.A.H_000909500; *PEPCK*, TcIL3000.A.H_000300300) by dsRNAs of 493–707 bp consistently reduced mRNA levels, but only to 30–60% of parental or non-induced levels (Fig 5E and 5F).

Given that T7-driven transcription in TcoSM cells from minichromosomal loci is at least as strong (in comparison to pol II transcription) as seen in *T. brucei* bloodstream forms [21], the lower penetrance of RNAi in IL3000 *T. congolense* does not appear to be a feature of integration site. Indeed, levels of knockdown with p3T7-TcoV are considerably stronger than for inducible constructs integrated at the tubulin locus (without the apparent reduction of levels in non-induced cells), and as strong as for constitutively expressed hairpins introduced into the genome [13]. Instead, it is likely that there is an intrinsically lower effect size resulting from expression of double-strand RNA in *T. congolense*, at least for the genome reference strain.

## Inducible knockdown of essential genes in *T. congolense* bloodstream forms

To test if levels of RNAi knockdown in *T. congolense* could produce defects expected due to loss of essential proteins, we used p3T7-TcoV to knockdown genes encoding α-tubulin (e.g. TcIL3000.A.H_000560300; δ*tuba*) and clathrin heavy chain (TcIL3000.A.H_000768200; δ*chc*). Knockdown of these genes produce distinctive 'FAT' and 'BigEye' phenotypes when targeted in *T. brucei* [35,36]. Consistent with RNAi against non-essential genes, RNAi reduced *CHC* mRNA levels to ~30% of parental (Fig 5E). Both δ*tuba* and δ*chc* produced the expected morphological change in ~50% of cells in independent clonal populations by 48 h post-induction (56% and 54% for δ*tuba* and δ*chc*, respectively; Fig 6A and 6B), similar to proportions of cells with morphological defects seen previously in inducible knockdown of α-tubulin in procyclic-form *T. congolense* [28]. Again, these data are compatible with an intrinsically lower penetrance of RNAi in IL3000 compared to *T. brucei* strains. They also suggest that there may be some attenuation of RNAi effect at later time points–as seen in a reduction of δ*chc* cells with an detectable morphological defect at 72 h versus 48 h (Fig 6B; mean 35% and 54%, respectively) and a slight increase in mRNA levels for all genes followed over 72 h induction (Fig 5F).

Induction of RNAi against either *TUBA* or *CHC* caused a growth defect in all clones analysed from 24 h post-induction (Fig 6C). However, even though defects were still apparent at later time points, in neither case did populations stop dividing completely–doubling every

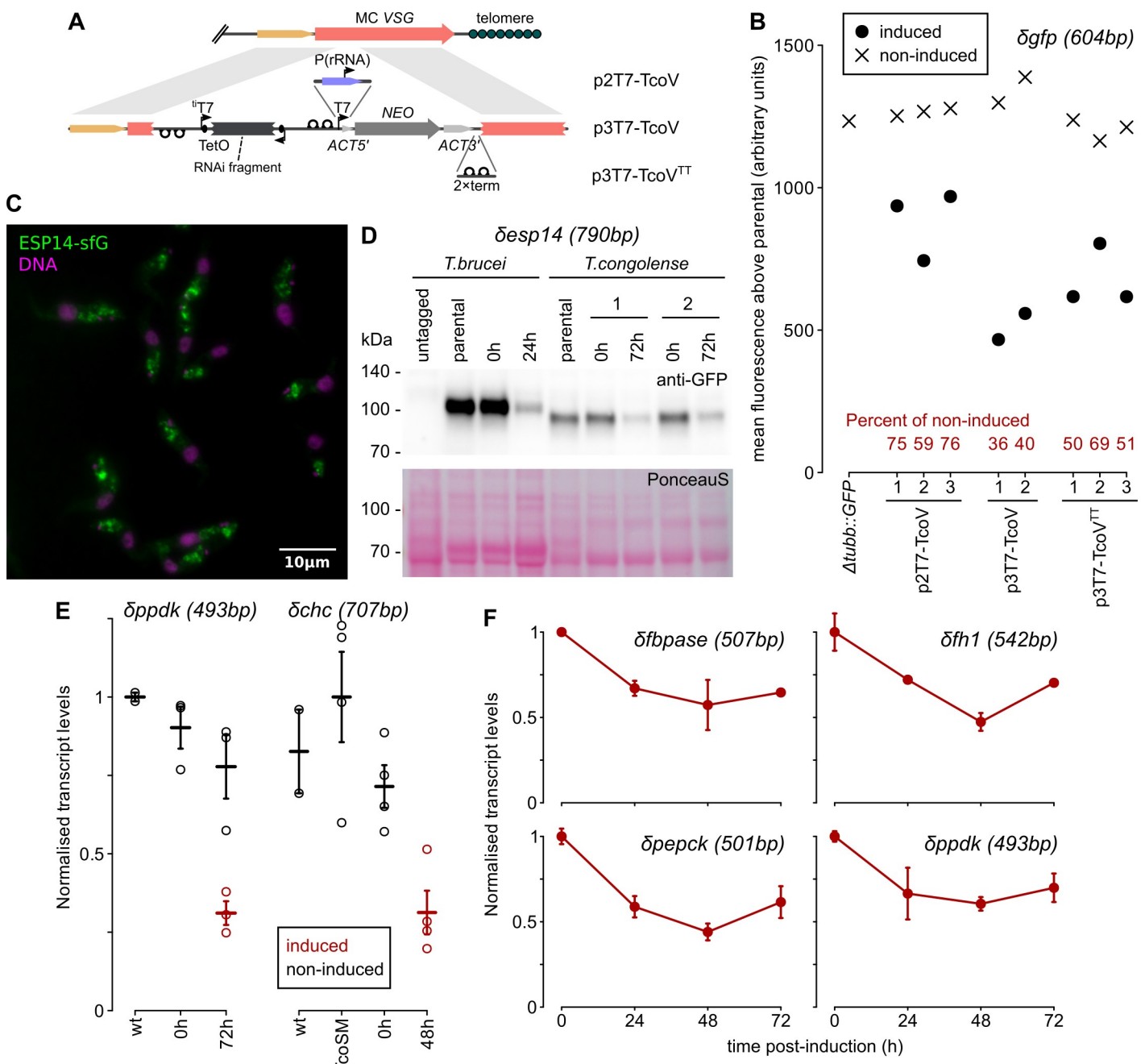

**Fig 5. Inducible RNA-interference from a silent minichromosomal locus in bloodstream-form *T. congolense*.** Numbers in parentheses following knockdown name refer to the total length of the gene fragment inserted between promoters in the RNAi construct. A) Schematic representation of RNAi construct architectures and integration. B) Quantification of knockdown of GFP expressed from the tubulin locus by RNAi targeting *GFP* (*δgfp*). Numbers on the x-axis represent independent clones. C) Localisation of the *T. congolense* orthologue of ESP14 (TcIL3000_7_180) by endogenous-locus tagging. Counter-staining of cells with DAPI (magenta) is also shown. D) Knockdown of endogenous locus-tagged ESP14 in bloodstream-form *T. brucei* and two independent clones of *T. congolense*. E) Quantification of target transcript levels following RNAi targeting pyruvate phosphate dikinase (TcIL3000.A.H_000922100, *δppdk*) or clathrin heavy chain (*δchc*) genes. Mean and SEM from 3 (*δppdk*) or 4 (*δchc*) biological replicates are shown along with individual measurements (dots). F) Time courses for target transcript levels following RNAi targeting 4 genes with putative metabolic roles. Bars: SEM from 3 biological replicates.

20.0 ± 1.8 h and 19.3 ± 2.0 h after 24 h induction for δ*tuba* and δ*chc*, respectively, compared to 10.3 ± 0.3 h for non-induced cells. Cells not producing a detectable morphological defect early in induction are not due to a substantial proportion of cells refractory to RNAi in the non-

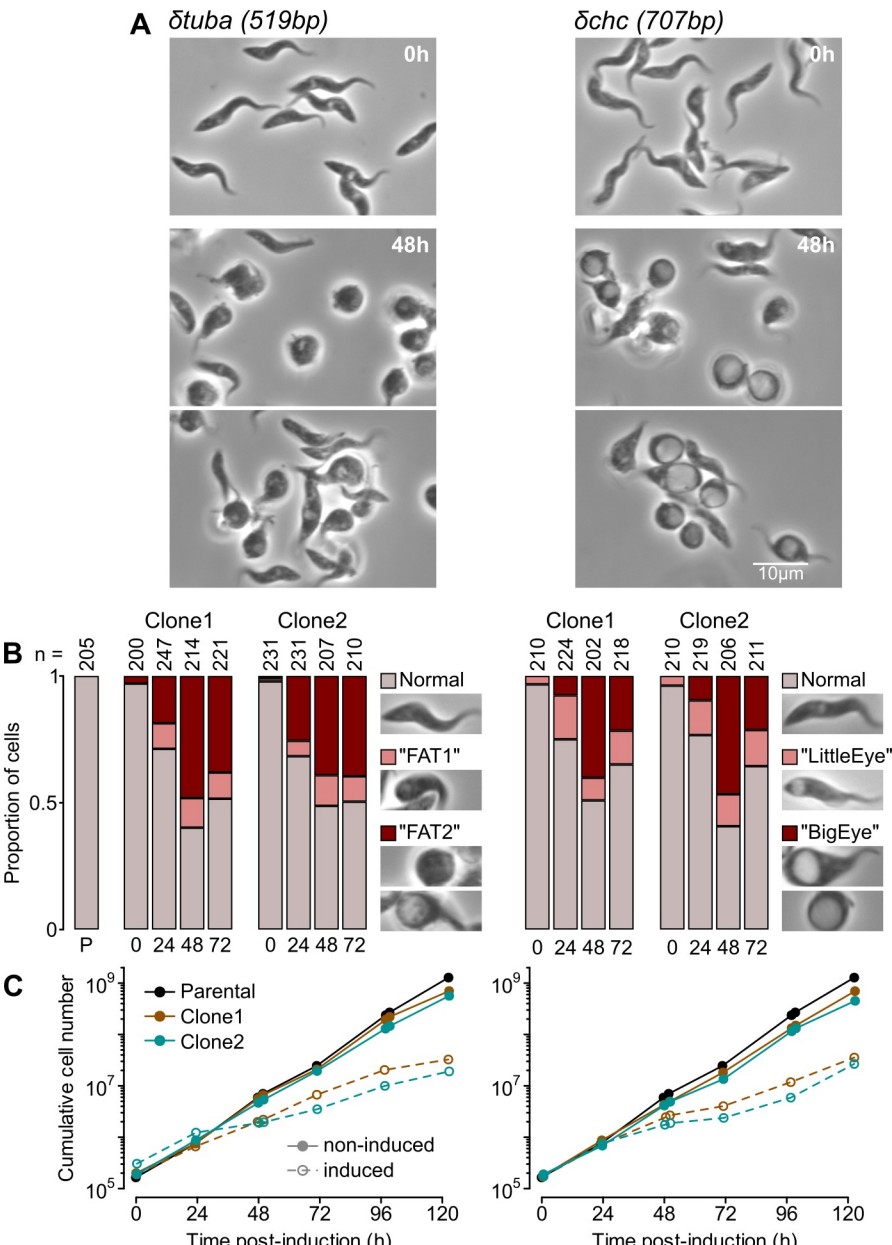

**Fig 6. RNAi targeting essential genes in *T. congolense* causes growth defects and morphological changes similar to those seen in *T. brucei*.** A) Phenotypic changes seen following knockdown of α-tubulin (*δtuba*) or clathrin heavy chain (*δchc*) in *T. congolense*. Images from 2 technical replicates of induction of a single clone (Clone 1) are shown. B) Quantification of gross morphological changes. Representative data from 1 of 2 biological repeats are shown. Numbers below bars are time post-induction (h); numbers above are cells analysed; P, parental (no RNAi construct). C) Growth of cells containing RNAi constructs in 0 (non-induced) or 1 μg ml$^{-1}$ (induced) tetracycline.

induced population, as they produce progeny that can be affected by knockdown. Instead, there appears to be a relatively stable proportion of the population where levels fall below a threshold necessary for apparent defect, at least in the short-term following induction. Importantly, this proportion was unchanged by freeze-thawing of clones, meaning that RNAi lines can be stored and re-analysed in future experiments (S4 Fig). However, there was some attenuation of RNAi defects after growth in culture without induction for 8 weeks (>120 generations; S4 Fig),

suggesting that there is some long-term pressure imposed by the presence of the RNAi construct–even without induction.

## Inducible double-strand break creation in *T. congolense* minichromosomes

A key technical development in the evolution of *T. brucei* genetic modification was the demonstration that double-strand breaks at specific loci could hugely increase the efficiency and specificity of transgene integration [15]. This is a critical component of high-complexity library approaches such as genome-scale RNA-interference target sequencing (RIT-Seq; [16]). The technology as currently applied in *T. brucei* involves introducing an I-SceI meganuclease recognition sequence at the target locus followed by transient expression of I-SceI to induce cutting. To implement a similar approach at *T. congolense* minichromosomes, we designed and built constructs to introduce an inducible, self-excising gene encoding I-SceI (with a monopartite nuclear localisation sequence from SV40 Large T-antigen) at the silent minichromosomal *VSG* (Fig 7A).

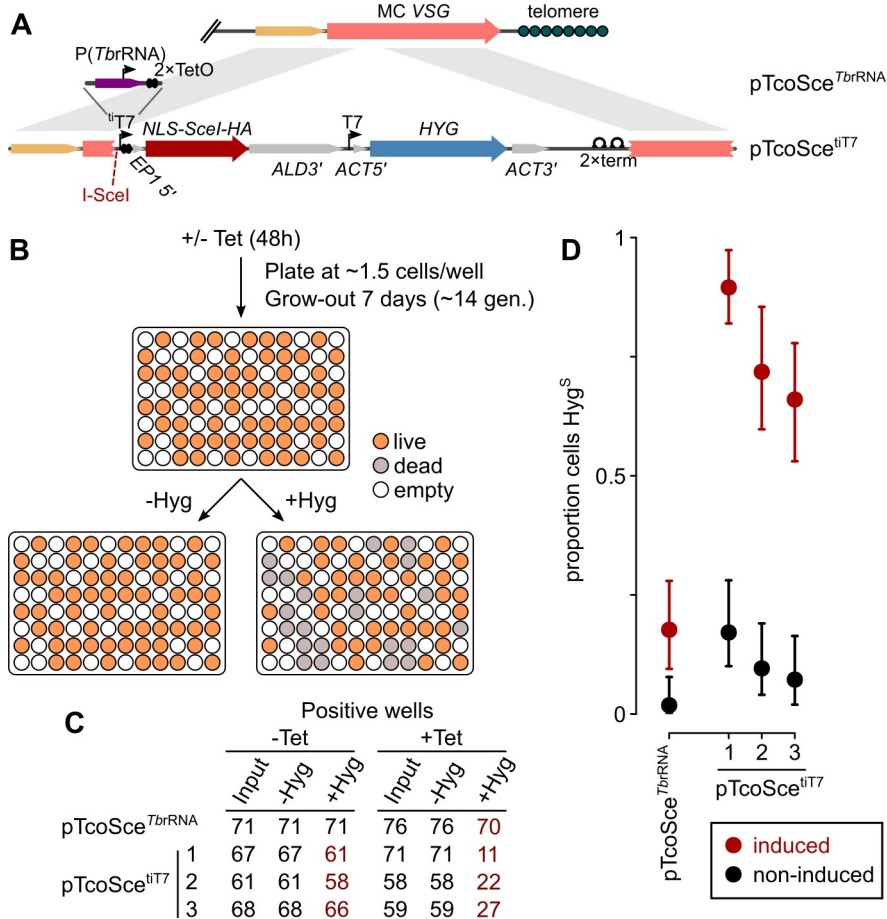

**Fig 7. Production of inducible, self-excising locus on *T. congolense* minichromosome.** A) Schematic representation of construct architectures and integration. B) Experimental set-up for estimation of locus loss. Tet, tetracycline; Hyg, hygromycin; P(*Tbr*RNA), *T. brucei* rRNA gene promoter. C) Number of positive wells on input plate and replicas in the absence (-Hyg) or presence (+Hyg) of hygromycin. D) Inferred proportion of hygromycin-sensitive cells following induction. Three independent clones are shown for pTcoSce^tiT7 (1–3). Bars: 90% confidence intervals for estimates from individual biological replicates.

In *T. brucei*, the published approach targeting a rDNA spacer has the disadvantage that leakiness in I-SceI expression means that cells containing the I-SceI construct (known as Sce* cells) are unstable on freeze-thaw and also during medium-term culture [37]. Although our data suggest levels of transgene expression from the targeted locus are very low in the absence of induction (Fig 4), it is not known what level of protein is necessary to produce cutting in a significant proportion of cells. To accommodate differences in I-SceI threshold, constructs were designed with the gene under the control of 2 alternative promoters (both controlled by two TetO elements): a T7 promoter (pTcoSce<sup>tiT7</sup>) expected to produce very strong expression, and a heterologous promoter for the rRNA genes from *T. brucei* (pTcoSce<sup>TbrRNA</sup>). The latter is highly dissimilar to the endogenous rRNA promoter and our data suggest it results in very low transgene expression when used in *T. congolense*, but is known to be well regulated in *T. brucei* [25].

Cutting of I-SceI in cells transfected with either pTcoSce<sup>tiT7</sup> or pTcoSce<sup>TbrRNA</sup> would cause loss of telomere-proximal *SceI* and *HYG* (Fig 7A). To test for the efficiency of double-strand break induction, we grew cells for 48 h in the presence or absence of tetracycline, plated without selection and then applied drug selection to test for loss of *HYG* (Fig 7B). Cells containing I-SceI controlled by heterologous *Tb*rRNA promoter presented no evidence of *HYG* loss in the absence of tetracycline but also few hygromycin-sensitive wells on induction (0/71 and 6/76, respectively, equating to a probability of loss in individual cells of $\leq 0.08$ and $0.1–0.3$; p = 0.03, Fisher's exact test; Fig 7C). In contrast, induction of T7-driven I-SceI expression resulted in production of hygromycin-sensitive wells at a rate equivalent to 65–92% of cells losing *HYG* across 3 independent clones (Fig 7C and 7D). Although the number of sensitive wells produced by non-induced lines was also significantly higher with this strong promoter compared to heterologous rRNA promoter (11/196 and 0/71, respectively; p = 0.04), the estimated loss for even the highest clone is equivalent to a loss in ~3% of cells per generation (17% over 48 h), which is sufficiently low for short-term growth without selection.

## Inducible double-strand break creation greatly increases transfection efficiency

To test if double-strand break induction increases transfection efficiency in *T. congolense*, we took TcoSM cells transfected with pTcoSce<sup>tiT7</sup> or pTcoSce<sup>TbrRNA</sup>, induced I-SceI expression for 16 h and then introduced a *T. congolense* RNAi plasmid that targets the Sce-modified locus (p3T7-TcoV containing a fragment of *TUBA*; Fig 8A). I-SceI induction time was chosen as being the approximate peak in specific ssDNA production seen in *T. brucei* [15] and also for simple convenience in experimental set-up. It is possible that other lengths of induction would further change efficiency, but this was not investigated here.

Induction of I-SceI controlled by heterologous rRNA promoter had no impact on transfection efficiency, which was indistinguishable from transfection of construct into TcoSM cells (~250 independent clones per transfection; Fig 8B). Induction of T7-driven expression, however, increased efficiencies to >10,000 clones per transfection (~50-fold above efficiency without cutting). For 2 of 3 independent clones there was evidence of considerable cutting even without tetracycline (Fig 8B). Given the very low background transcription for tetracycline-inducible transgenes at this minichromosomal locus, this seems anomalous–particularly as there was no corresponding loss of *HYG* in these clones during normal growth (Fig 7D). Our current hypothesis is that this reflects stochastic displacement of TetR in a proportion of cells as a result of electroporation stress. This would explain the high variability in the effect (100-fold between experiments and clones; Fig 8B). Fortunately, it is of no practical disadvantage, as induced efficiencies are consistent and high across clones. Significantly, all clones were

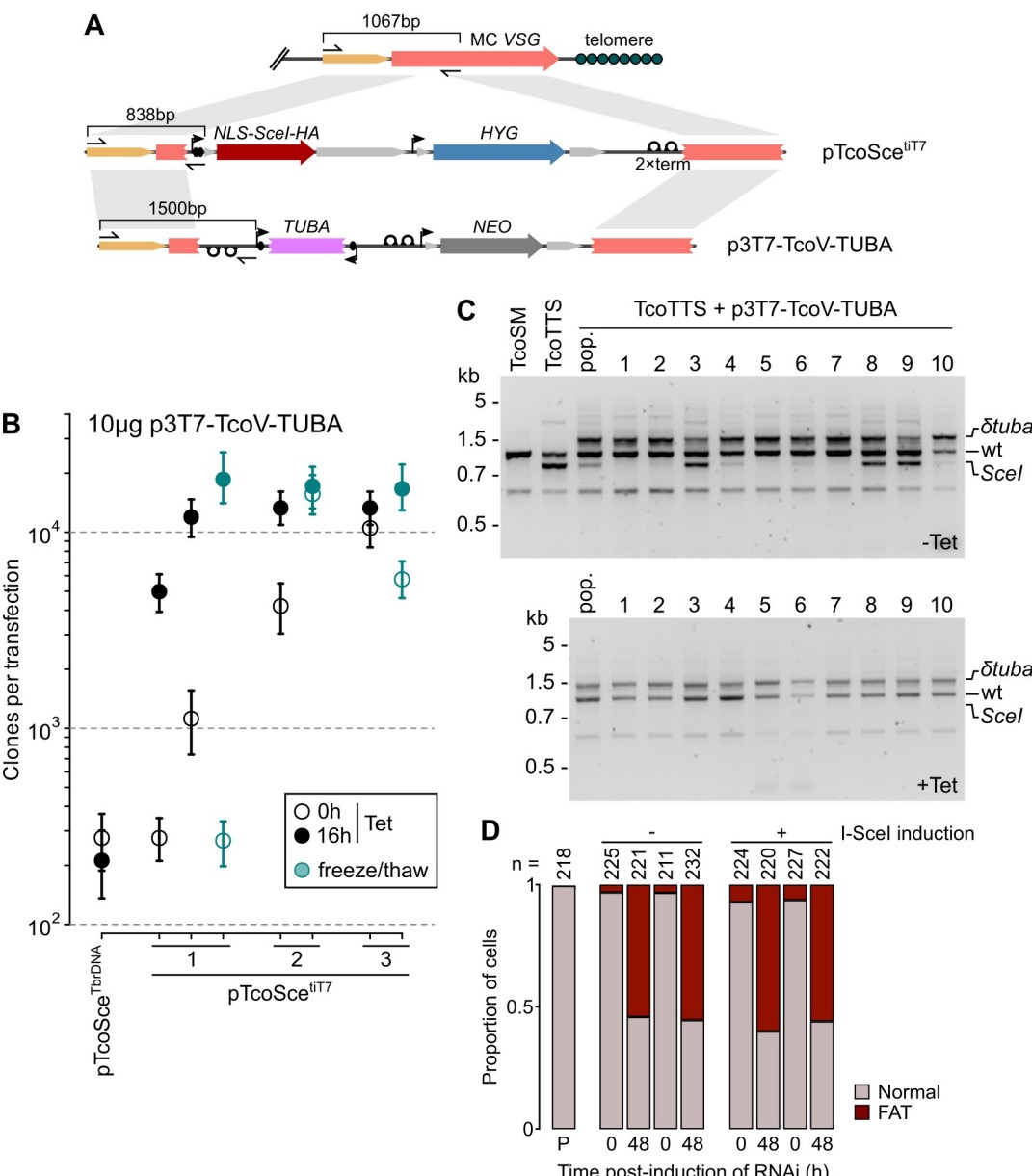

**Fig 8. Increased efficiency of integration at induced double-strand break locus.** A) Schematic representation of integration events and resultant unique amplicon sizes. The targetted locus, TcIL3000.A.H_000093600, is first modified by integration of pTcoSce[tiT7] or pTcoSce[TbrDNA] and may be subsequently replaced by p3T7-TcoV-TUBA, which has the same targeting sequences. B) Efficiency of stable transfection following induction of double-strand break in 3 independent clones carrying self-excising locus at minichromosome *VSG*. Independent replicates of transfection are shown for each clone. Bars: 90% confidence intervals for estimates in individual transfections. Clone 1 was selected as the progenitor of the TcoTTS cell line. Freeze/thaw indicates samples that have been stored in liquid nitrogen and defrosted prior to transfection. C) Multiplex PCR showing modification of minichromosome *VSG* loci following introduction of a construct targeting the *VSG* (p3T7-TcoV-TUBA) after 0 h (-Tet) or 16 h (+Tet) induction of double-strand break at the target locus in TcoTTS cells. Distinguishable amplicons are produced from unmodified loci (wt), and loci modified by either pTcoSce[tiT7]/pTcoSce[TbrDNA] (*SceI*) or p3T7-TcoV-TUBA (*δtuba*). D) Production of phenotypic changes due to knockdown of α-tubulin (*δtuba*) in 2 independent clones produced from TcoTTS cells either without (-) or with (+) induction of double-strand break during transfection.

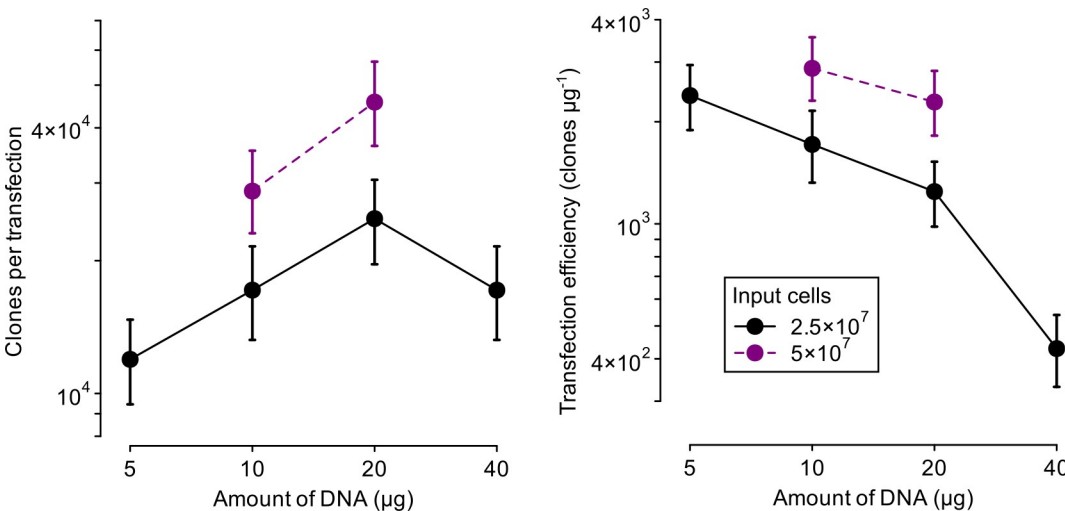

**Fig 9. Optimization of transfection efficiency in TcoTTS cells following induction of double-strand break.** A single purified preparation of p3T7-TcoV-TUBA was used for all transfections. Bars: 90% confidence intervals for estimates in individual transfections.

also stable to freeze-thawing of cultures (Fig 8B) and could be maintained in culture for at least 3 weeks without loss of high-efficiency.

We selected the clone with greatest fold-change in transfection efficiency as the basis for a stable library-recipient cell line. This cell line, based on IL3000 TcoSM cells, is named TcoTTS for the presence of T7RNAP, TetR and inducible I-SceI. TcoTTS cells are compatible with both our RNAi and over-expression constructs targeting *VSG* TcIL3000.A.H_000093600, and are suitable for generation of high-complexity libraries. Optimization of transfection parameters in TcoTTS cells following frozen storage showed that this line can routinely and consistently produce >40,000 independent clones per transfection (using 20 μg input DNA and $5x10^7$ induced cells; Fig 9).

Although the targeted *VSG* (TcIL3000.A.H_000093600) is present in the long-read IL3000 assemblies as a single copy [5], analysis of integration events shows at least 3 copies in IL3000 cells, since TcoTTS cells contain both modified and unmodified TcIL3000.A.H_000093600 loci and introduction of p3T7-TcoV without induction of I-SceI can either replace pTcoSce$^{tiT7}$ (e.g. clone 1 in Fig 8C) or modify a *VSG* copy (e.g. clone 3 in Fig 8C) and still leave ≥1 unmodified copy. Copies of TcIL3000.A.H_000093600 likely represent multiple copies of the minichromosome itself that could not be distinguished during assembly. As expected, induction of I-SceI pushes all integration events to the single modified locus (Fig 8C), meaning that all integration events in induced TcoTTS are modified at precisely the same genomic location. Importantly, phenotypes associated with δ*tuba* could be induced equally in cell lines derived by I-SceI induction as those derived without induction (Figs 6B and 8D), showing that the process does not interfere with generating RNAi against essential genes.

## Genetic modified IL3000 cells for in vivo disease models

In order to use directly modified bloodstream-form *T. congolense* in animal models of disease, it is necessary that cells following selection still establish infections with reasonable dynamics. It would also be beneficial to generate forms of IL3000 that can be followed through an infection by in vivo imaging. To this end, we designed and built constructs to stably express firefly luciferase using an endogenous rRNA promoter (Fig 10A), in a manner analogous to those

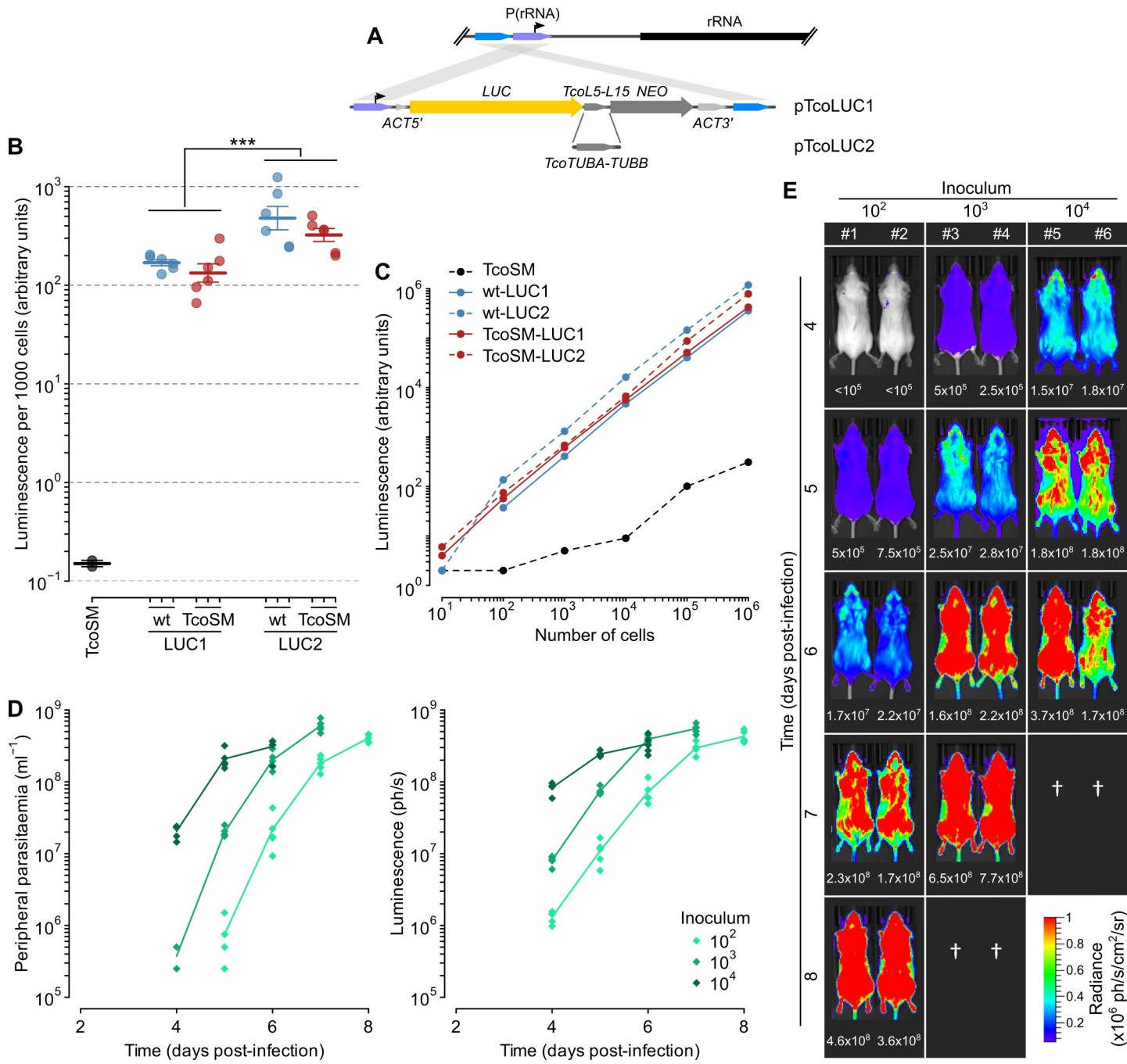

**Fig 10. Transgenic *T. congolense* bloodstream lines for visualisation of in vivo infection.** A) Schematic representation of construct architectures and integration. Constructs differ only in *T. congolense* intergenic region between CDSs in transgenic bicistron. B) Luminescence in wild-type (wt) or TcoSM cells stably transfected with pTcoLUC1 or pTcoLUC2. Technical duplicates from 3 independent clones of each line are shown along with mean and SEM across clones. C) Linearity of luminescence in the highest-expressing wild-type (wt) or TcoSM LUC1/2 clones. D) Infection dynamics of *T. congolense* LUC2 cells in a mouse model of AAT monitored by bioluminescence and parasitaemia in peripheral blood (5 animals for each inoculum). Inoculum refers to total dose of parasites used to infect animal at time 0. E) In vivo imaging of infection density in mice inoculated with *T. congolense* LUC2 cells. Numbers below images of individual mice are estimates of the peripheral parasitaemia (cell ml$^{-1}$) made by haemocytometry. Data from 2 representative animals for each inoculum are shown.

used in *T. brucei* [38] and *T. vivax* [39]. Integration is duplicative, such that transcription of the downstream *RRNA* polycistron is not disrupted and the modifications carry no bacterial DNA to the integration site. Two alternative endogenous intergenic regions from highly expressed *T. congolense* genes (between the genes encoding L5 and L15 ribosomal proteins, and *TUBA-TUBB*) were placed at the 3' end of the luciferase CDS. Integration of either construct in IL3000 wild-type or TcoSM cells produced detectable luciferase at levels >1000-fold above background (Fig 10B), but levels associated with the tubulin intergenic (LUC2) were significantly higher (p < 0.001; Student's t-test). Luminescence in vitro is linear across >5-orders of magnitude and as few as 10 cells in total are detectable above background (Fig 10C).

To create a model of acute trypanosomiasis, $10^6$ modified LUC2 cells were injected into BALB/c mice via the intraperitoneal route. First parasitaemic peak was seen at day 4 post-infection, demonstrating that modified cells are still infectious directly from culture. Cells from the second parasitaemic wave were further passaged twice in BALB/c mice, each to the first parasitaemic peak. Luciferase expression was maintained across this in vivo growth in the absence of selection and infections with this line result in a predictable dose-dependent time to first peak parasitaemia across at least 2-orders of magnitude (Fig 10D). Infection dynamics are also very similar in male and female mice and in an albino strain derived from *C57BL/6* (S5 Fig). Monitoring parasites by in vivo bioluminescence imaging shows linear doubling of parasite numbers during exponential growth proportional to the initial dose and with an estimated in vivo doubling time of 8.4 h. However, counts of parasites in peripheral blood appear to substantially underestimate total *T. congolense* abundance at low parasitaemia (Fig 10D)–probably due to the vascular adherence of parasites that is a feature of *T. congolense* infections [40]. Distribution of *T. congolense* in this acute model extends widely through the animal (Fig 10E) and there is no strong evidence for tissue-specific accumulation as is seen at later stages of infection with *T. brucei* [38] or *T. vivax* [41].

## Discussion

### Transfection of bloodstream-form IL3000 cells

Here we have described, for the first time routine, direct genetic modification of bloodstream-form *T. congolense* IL3000. Transfection efficiencies when targeting endogenous loci are good (100–4000 independent clones per transfection)–at least as high as for Lister 427 bloodstream-form cells, which are the most commonly modified *T. brucei* strain–and do not require heterologous DNA modifiers, such as recombinases, restriction endonucleases or Cas9. While this is a great benefit for *T. congolense* research, it does raise the question as to why previous attempts at stable transfection proved so challenging.

The IL3000 line used herein is a descendant of the original in vitro bloodstream-form line [14] and is still infective in animals. Our transfection procedure is covered in detail in Materials and Methods, but is also only modestly different from standard short-pulse electroporation ('nucleofection') procedures used previously–most of which gave no stable transfectants. Although not systematically addressed, our experience suggests that bloodstream-form *T. congolense* cells are much more sensitive to bacterial endotoxin than *T. brucei* and high-purity DNA appears to be an absolute requirement for high efficiency stable transfection. We only perform transfection with DNA purified by anion exchange and transfection buffer made from high-purity stocks.

Notwithstanding influence from the above, the major determinant of previous difficulties in genetic modification of *T. congolense* is likely to be simply that the constructs were not designed for this organism, but were heterologous application of *T. brucei* constructs. Many genomic elements definitely cannot be transferred in this manner: the *T. brucei* procyclin

promoter is not functional in *T. congolense* [27], and our data suggest that the *T. brucei* rRNA promoter has very limited activity. Moreover, while it is clear that at least some *T. brucei* intergenic regions are processed sufficiently in *T. congolense* to produce detectable levels of protein [14,17,28], the efficiency of processing compared to native sequences is unknown and likely to be much lower in many cases. More significantly, the majority of reported transfection attempts have also used heterologous *T. brucei* sequence for targeting integration [13,14,17]. It is unknown how transfection efficiency scales with identity of targeting sequence in *T. congolense*, but in *T. brucei* comparable levels of mismatch (10% for *TUBB*) would cause a 50-fold decrease in efficiency [42]. This may not explain all the improvement in efficiency we have achieved (perhaps as much as 1000-fold), but likely provides a substantial contribution. Importantly, this implies that, although the work here is focused on bloodstream-form cells, all the tools developed will also greatly facilitate work using other replicative stages. None of our *T. congolense* constructs contain stage-specific elements and we have shown that procyclic-form cells can be modified with similar ease to bloodstream forms (Fig 1)–with approximately 500-fold greater efficiency than previous studies (>50 and ~0.1 independent clones per million cells, respectively; [17]).

## RNAi effectiveness in *T. congolense* IL3000

Tetracycline-regulated T7RNAP-driven transcription from minichromosomal loci in TcoSM cells is at least as strong with respect to pol II transcription as in *T. brucei* SMB and 90-13 lines [21], but T7-based RNAi in *T. congolense* had consistently lower effect size compared to typical *T. brucei* knockdowns. To our knowledge, there are no published systematic tests of the effect of dsRNA length on knock-down effectiveness in *T. brucei*, although fragments as small as 50–60 bp have been reported to reduce endogenous mRNA levels when expressed in cells [36,43]. In the absence of strong evidence for length effects, most knockdowns using head-to-head promoter arrangements in *T. brucei* employ fragments in the range 250–800 bp. In our *T. congolense* system, gene fragments in the range ~500–800 bp reduced mRNA and protein levels to ~30–50% of parental levels with no evidence for length effects. It is possible that much longer dsRNA could increase RNAi effectiveness in *T. congolense*, but current data suggest this is unlikely. Moreover, similar levels of knockdown were seen previously when expressing RNA hairpins from constitutively active loci [13] and our attempts to improve RNAi effect size using inducible single-promoter hairpins gave no greater phenotype penetrance than head-to-head constructs. These data imply that *T. congolense* and/or the IL3000 strain have an intrinsically lower RNAi effect size than commonly tested *T. brucei* strains. This does not appear to be due to differences in expression of the core RNAi machinery, as levels of mRNA encoding the trypanosome homologues of Argonaute [44,45], Dicer [46,47] and the RNA Interference Factor 4 (RIF4) are slightly higher in bloodstream-form *T. congolense* compared to *T. brucei* (Fig 11A). The encoded proteins are also well conserved within African trypanosomes and no more divergent in *T. congolense* than would be expected from lineage history (Fig 11B).

Importantly, although RNAi effectiveness–both in terms of proportion of cells exhibiting a defect and the defect severity–is lower than in *T. brucei*, knockdown is sufficient to produce gross morphological defects in ~50% of cells when targeting two essential genes. This effect on only part of a clonal population does not appear to be due to a high proportion of refractory cells. It appears instead to be a threshold effect, and progeny of cells that had no detectable morphological change can go on to develop defects. As a result, the corresponding halving of the population growth rate after 24 h RNAi induction is persistent, at least over a few days, and a large difference in total cell numbers can thus accumulate (~20-fold difference by 5 days post-induction). This means that, even in the absence of dramatic cell death, such RNAi-

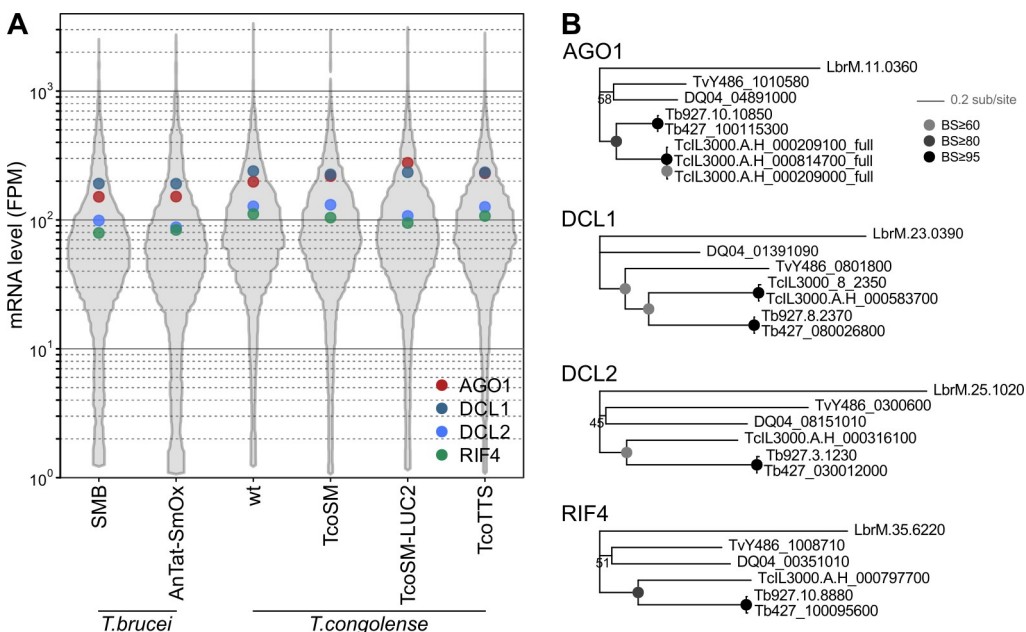

**Fig 11. Expression and conservation of components of the RNAi machinery in T. brucei and T. congolense.** A) Total mRNA levels (fragments per million; FPM) for AGO1, DCL1, DCL2 and RIF4 homologues in bloodstream-form cells. For comparison, the distribution of FPM for all detected genes is also shown (grey). RNA levels for the 3 near-identical AGO1 genes in *T. congolense* (>99% identical at the nucleotide level and indistinguishable across most of their length in RNA-Seq) have been summed to reflect the total levels of mRNA for the protein. B) Maximum likelihood phylogenies for homologues of components in *Leishmania braziliensis* (LbrM), *Trypanosoma grayi* (DQ4), *T. vivax* (TvY486), *T. congolense* (TcIL3000) and *T. brucei*. Sequence for both Lister 427 (Tb427) and TREU927/4 (Tb927) strains are shown for *T. brucei*. *T. congolense* gene models suffixed "_full" have been corrected to start at the unannotated in-frame ATG, which in all cases produces an uninterrupted CDS >99% identical to either TcIL3000_10_9140 or TcIL3000_10_9150 in the short-read assemblies. Phylogenies were inferred from aligned protein sequences using FastTree 2.1.9 ([60]; LG substitution matrix, gamma-distributed variation in substitution rate approximated to 12 discrete categories). Node support is derived from 100 bootstrap replicates.

induced changes are readily amenable to identification by high-throughput methods relying on differential cell growth such as RIT-Seq (for example, the fold-change in *TUBA* fragments by day 6 of RIT-Seq in *T. brucei* is also ~20-fold; [16]). This is particularly relevant given that we have also developed here a system for production of high-complexity libraries in *T. congolense* analogous to Sce* cells used in *T. brucei* [15].

## A toolkit for functional genetics in *Trypanosoma congolense*

Work in *T. brucei* (and then only in a limited set of strains) dominates functional genetics in African trypanosomes. This is in spite of *T. congolense* and *T. vivax* being the more important parasites for AAT and it being known that the species differ in significant aspects of their biology. The dominance of *T. brucei* work is due in large part to the wealth of tools for functional genetics built up across more than 3 decades of work. Here we have attempted to substantially accelerate work in *T. congolense* by developing a basic set of tools and methods that open the organism up to functional genetics–designing each tool from first principles to suit the *T. congolense* genome.

With homologous targeting sequences, we have shown that modification of endogenous loci for tagging or gene knockout is routine in *T. congolense*, even for bloodstream stages. To date, we have only used 3 selective drugs (hygromycin, G418 and puromycin), but bleomycin has been used elsewhere [14,28] and additional selective drugs used in *T. brucei* are highly

likely to be usable. Development of a well-regulated system for inducible ectopic (over-)expression was a key early tool in *T. brucei*. We have described a *T. congolense* single marker line (TcoSM) that expresses codon-optimized TetR and T7RNAP using *T. congolense*-specific processing sequences. Combining this line with the use of minichromosomal loci as stable, silent targets for inducible transgenes brings levels of expression in the absence of induction from only ~4-fold below induced as previously achieved [14] to below the level of detection (>700-fold regulation), creating a system suitable for toxic gene expression or conditional knockouts. We have also designed a *T. congolense*-specific RNAi construct, p3T7-TcoV, that integrates at a silent minichromosomal *VSG* and validated it against multiple genes. This is by no means the first application of RNAi in *T. congolense*, which goes back nearly two decades [28], but should greatly increase the utility of a technology that has been so important in analysis of gene function in *T. brucei*. To date, we have seen no evidence of significant clonal variation in knockdown at the mRNA, protein or phenotypic levels, suggesting that we can anticipate good reproducibility of RNAi using p3T7-TcoV. In addition to this, genetically modified IL3000 bloodstream-form cells are still infectious and the production of a stable, highly-luminescent line provides a tool for vaccinology or other work requiring in vivo disease models. Since modified *T. congolense* minichromosomes have good stability in the absence of selection, this system also allows regulation of transgene expression or RNAi during an infection.

The tools above enable work analogous to the majority of studies in *T. brucei* to be performed directly in bloodstream-form *T. congolense*. Since specific modification of endogenous loci in bloodstream-form *T. congolense* is now routine, we have not prioritized testing of CRISPR/Cas systems in the first instance, but such a system would be a useful future extension to allow modification of multiple alleles/loci. However, some of the most transformative technologies in *T. brucei* in recent years depend on the production of not individual mutants, but libraries containing many thousand mutants (e.g. RIT-Seq [16,37]). The TcoTTS line (T7RNAP, TetR and inducible I-SceI) allows high-complexity library production in *T. congolense* for the first time, robustly giving >40,000 independent clones per transfection with p3T7-TcoV. Importantly, this is sufficient efficiency for genome-scale applications encompassing several mutants for each gene in the genome. The stability of TcoTTS with respect to storage and at least medium-term growth is an additional convenience in comparison to the current technology in *T. brucei* as it removes the need for fresh derivation before each experiment. Moreover, since all copies of the locus appear to behave indistinguishably with respect to transgene expression/RNAi, this technology can be trivially recreated in other parental lines without the necessity of screening multiple clones.

The work here is focused on bloodstream-form *T. congolense*, as the lifecycle stage of highest experimental interest. However, all the constructs developed are applicable without modification in other stages–an application for which *T. congolense* is well-suited due to the ability to reproduce its whole lifecycle in vitro [14]. It also provides an example of the application of genome and transcriptomic sequence information, cost-effective DNA engineering tools, and information from decades of work in *T. brucei* for the rapid development of functional genetics in other important disease-causing trypanosomes. Application of this to other lines or species is relatively easy, but can only be applied to cells that can be grown as stable in vitro cultures.

## Materials and methods

### Ethics statement

Animal experiments were performed under UK Home Office regulations (project licences PD3DA8D1F and P98FFE489) and European directive 2010/63/EU. Research was ethically approved by the Sanger Institute Animal Welfare and Ethical Review Board.

## Cell culture and transformation

Bloodstream-form *T. congolense* IL3000 strain, originally isolated at the International Live-stock Research Institute (Nairobi, Kenya) and derived from infected mouse blood by [14], was a kind gift of Prof. Mike Barrett (University of Glasgow, UK). Cells were cultured at 34˚C and 5% $CO_2$ in TcBSF1 without red blood cells [14], with the modification that only 15% goat serum (GIBCO) was used and no serum-plus. Procyclic-form *T. congolense* IL3000 were a kind gift of Prof. Wendy Gibson (University of Bristol, UK) and were cultured in unvented flasks at 28˚C in TcPCF3 [14]. Bloodstream-form *T. brucei* was grown in HMI-9 medium supplemented with 15% fetal bovine serum at 37˚C and 5% $CO_2$ [48].

2-3x$10^7$ cells and 10 μg of cut plasmid were used unless otherwise stated in the text. DNA was concentrated for transfection by ethanol precipitation and resuspension in 30 μl Tb-BSF (90 mM sodium phosphate, 5 mM KCl, 0.15 mM $CaCl_2$, 50 mM HEPES, pH 7.3; [49]). Actively-dividing, healthy cells (at ~2x$10^6$ and ~2x$10^7$ cells ml$^{-1}$, for bloodstream- and procyc-lic-form trypanosomes, respectively) were harvested by centrifugation at 1200 $g$ for 10 min, washed in a small volume of Tb-BSF, re-pelleted at 1200 $g$ for 2 min, and resuspended in 90 μl Tb-BSF. Cut DNA in Tb-BSF was added to these suspensions (final volume of 120 μl) and cells were electroporated in 2 mm cuvettes using an Amaxa Nucleofector 2b device (Lonza) with program 'Z-001'. Cells were quickly transferred to flasks containing pre-warmed medium and allowed to recover for 8 h. After recovery, selection was applied by addition of antibiotics to final concentrations of 0.5 μg ml$^{-1}$ puromycin, 0.2 μg ml$^{-1}$ G418, or 0.5 μg ml$^{-1}$ of hygromycin B. For pGad constructs (containing a single GFP and drug polycistron), selection was carried out in the presence of induction with 1 μg ml$^{-1}$ tetracycline.

For transfection of TcoTTS cells with induction of I-SceI expression, the same procedure was followed as above, except that selective pressure for background (hygromycin B) was removed from cells 48 h prior to electroporation, and 1 μg ml$^{-1}$ tetracycline added 16 h prior to electroporation.

To estimate transfection efficiency, samples of cells immediately following recovery after electroporation were diluted in fresh, selective medium and distributed across 96 well plates. The proportion of total population plated was selected based on expected transfection effi-ciency, but for routine transfections 20% and 4% of each transfection was plated on 2 indepen-dent plates, and 2% and 0.4% for induced TcoTTS cells (providing accurate estimates of efficiency in ranges 100–6000 and 1000–60000 independent clones, respectively). Numbers of independent transfectants were derived from the number of positive wells based on the expected Poisson-distributed filling of wells. Confidence intervals of estimates were derived from the distribution of quantiles of $10^6$ simulated platings at 0.001 to 10 cells well$^{-1}$.

Induction of RNAi or ectopic transgene expression was by addition of 1 μg ml$^{-1}$ tetracycline to culture medium.

## Design and construction of plasmids

Sequences and graphical maps for all new constructs used in this work are available at www.wicksteadlab.co.uk and www.catarinagadelha.com, and also in S1 Dataset. All primers used for amplification of *T. congolense* targeting sequences are provided in S1 Table.

For tagging of proteins by modification of endogenous loci, constructs were derived from pEnNY0, pEnNmSt0-N [50] and pEnNmSc0-N [51] in the case of N-terminal tagging, or pSiS-HHsfG [34] for C-terminal tagging. In all cases, 300–800 bp of targeting sequence from each side of the integration site was amplified from *T. congolense* IL3000 genomic DNA and incorporated into the constructs using standard methods (see S1 Table). pGad10-TcoTUB, for knockout of a *TUBB* allele in *T. congolense*, was derived from pGad9 [26] by replacement of 5'-

and 3'- end of the *GFP-HYG* polycistron (without the promoter) by *TUBA-TUBB* and *TUBB--TUBA* intergenic regions, respectively (see S1 Dataset).

Since codon usage bias is very similar between *T. brucei* and *T. congolense*, CDS encoding TetR and T7RNAP for pTcoSM were taken from pSmOx ([22]; kindly provided by Steve Kelly, University of Oxford). All intergenic and targeting regions were amplified from *T. congolense* IL3000 genomic DNA, and the construct built by Gibson assembly in a pBluescriptII-SK (+)-derived backbone (see S1 Dataset).

Constructs for ectopic inducible expression of GFP from minichromosomal loci were derived from pGad9 [26] by incorporating 300–800 bp targeting sequences from each side of the integration sites amplified from *T. congolense* IL3000 genomic DNA. Amplicons of 369 bp repeat were size-selected for 2 tandem repeats (1 repeat at each end of integration site) by excision from agarose gel. p2T7-TcoV was derived from p2T7$^{Ti}$ [33], pEnNmSt0-N [50] and amplicons from *T. congolense* IL3000 genomic DNA by Gibson assembly (see S1 Dataset). This was subsequently modified to create p3T7-TcoV and p3T7-TcoV$^{TT}$ by standard means.

To generate a self-excising construct that creates double-strand breaks at minichromosomal *VSG*, sequence encoding I-SceI with an N-terminal nuclear localisation signal and C-terminal HA tag controlled by an inducible *T. brucei* rRNA promoter was obtained from pLew100:: NLS-ISceI-HA ([52]; kindly provided by Nina Papavasiliou, German Cancer Research Center, Heidelberg). This was combined by Gibson assembly with sequence elements from pGad9 and p3T7-TcoV to create pTcoSce$^{TbrRNA}$, which was subsequently modified by replacement of the inducible promoter by standard means to create pTcoSce$^{tiT7}$.

For stable expression of luciferase, CDS encoding firefly luciferase was taken from pLEW100 [21] and neomycin resistance cassette from p3T7-TcoV. These were assembled with amplicons from *T. congolense* IL3000 genomic DNA by Gibson assembly in a pBluescriptII-SK (+)-derived backbone.

All constructs were amplified in XL1 Blue *Escherichia coli* and purified using an anion-exchange column of the appropriate size (QIAGEN plasmid kits).

### Quantitative analysis of RNA

All RNA preparations were made from 6-10x10$^7$ actively growing cells using High Pure RNA isolation kit (Roche) and integrity of RNA assessed by denaturing agarose gel electrophoresis.

For RNA-Seq, sample preparation and sequencing was performed by University of Nottingham **Deep Seq facility.** RNA concentrations were measured using a Qubit Fluorometer and Qubit RNA BR Assay Kit (ThermoFisher Scientific) and integrity assessed using a TapeStation 4200 and RNA ScreenTape Assay Kit (Agilent). mRNA was purified from 1 μg of total RNA using the NEBNext Poly(A) mRNA Magnetic Isolation Module (New England BioLabs). Indexed sequencing libraries were then prepared using the NEBNext Ultra Directional RNA Library Preparation Kit for Illumina (New England BioLabs) and NEBNext Multiplex Oligos for Illumina, Index Primers Set 2 and Set 3 (New England BioLabs). Libraries were quantified using a Qubit Fluorometer and Qubit dsDNA HS Kit (ThermoFisher Scientific). Fragment-length distributions were analysed using a TapeStation 4200 and High Sensitivity D1000 ScreenTape Assay (Agilent). Libraries were pooled in equimolar amounts and final library quantification performed using the KAPA Library Quantification Kit for Illumina (Roche). The pool was sequenced using a NextSeq 500 System (Illumina) and mid-output 150 cycle kit v2.5 (Illumina), providing >12 million pairs of 75-bp paired-end reads passing filter per sample. http://www.ebi.ac.uk/ena

Reads were trimmed for quality and adapter using Trim Galore v0.4.4 (www.bioinformatics.babraham.ac.uk; -q 28—illumina—stringency 3—length 50) and aligned the

TriTrypDB *T. congolense* IL3000_2019 or *T. brucei* Lister 427_2018 assemblies (v46; based on the work of [5,53]) using bowtie2 v2.3.4 ([54];—no-mixed—no-discordant -I 50 -X 500). Total transcript abundance was assessed using HTSeq v0.5.4 [55]. Read depth analysis was performed using bedtools v2.25.0 [56].

For quantitative RT-PCR of transcripts in *δchc cells*, total RNA was reverse transcribed using avian myeloblastosis virus reverse transcriptase (New England BioLabs) primed by T15N. cDNA (or no-RT control) generated from 20 ng total RNA was used as template in each PCR reaction replicate. qPCR was performed on an Mx3000 system (Agilent) with a standard Taq polymerase mix supplemented with 0.5x EvaGreen dsDNA dye (Biotium). Target mRNA levels were normalised using levels of *PFR1* (TcIL3000.A.H_000346600) transcript. For quantitative RT-PCR in other RNAi cell lines, total RNA was reverse transcribed using a High Capacity cDNA RT kit (Applied Biosystems). qPCR reactions were carried out with SensiFAST SYBR Hi-ROX mix (Bioline), in a Rotor-Gene 3000 system (Corbett Research). Target mRNA levels were normalised using levels of *TERT* (TcIL3000.A.H_000960200, for other RNAi) transcript. Primers for *TERT* were as in [57], other primers used in quantitative PCR are provided in S2 Table. Samples were biological triplicates (for individual clones) or duplicates of 2 independent clones (*δchc*).

## Microscopy

For analysis of localisation of tagged proteins or soluble fluorescent protein by native fluorescence, cells were harvested from mid-log phase cultures, washed twice in phosphate-buffered saline (PBS; 137 mM NaCl, 3 mM KCl, 10 mM $Na_2HPO_4$, 1.8 mM $KH_2PO_4$) and allowed to settle for 5 min onto glass slides at ~$2x10^7$ cell ml$^{-1}$ density. Cells were fixed for 5 min in 2% (w/v) formaldehyde, permeabilised in -20°C methanol for 10 min, re-hydrated in PBS and incubated with 15 ng ml$^{-1}$ 4′,6-diamidino-2-phenylindole for 5 min, before mounting in 1% (w/v) 1,4-diazabicyclo[2.2.2]octane, 90% (v/v) glycerol, 50 mM sodium phosphate, pH 8.0. Counter-staining of cells expressing ESP14-sfGFP was by incubation with 25 µg ml$^{-1}$ tomato lectin conjugated to AlexaFluor 594 (Invitrogen) for 20 min. Images were captured on an Olympus BX51 microscope equipped with a 100x UPlanApo objective (1.35 NA; Olympus) and Retiga R1 CCD camera (Qimaging) without binning. All images of fluorescent proteins were captured at equal exposure settings without prior illumination. Images for level comparison were also processed in parallel with the same alterations to minimum and maximum display levels, except where stated. Image acquisition was controlled by µManager open source software [58]. Processing and analysis were performed in ImageJ [59].

## Immunoblotting

To test either tagged protein ablation by RNAi or ectopic gene expression, lysates from $2x10^7$ cells were separated by reducing SDS-polyacrylamide gel electrophoresis and electro-transferred onto nitrocellulose membrane (GE healthcare) in 25 mM Tris, 192 mM glycine, 0.02% SDS, 10% methanol. Membranes were blocked with 5% (w/v) skimmed milk in TBS-T (20 mM Tris-HCl, pH 7.5, 150 mM NaCl, 0.1% (v/v) Tween-20) and protein detected by either 800 ng ml$^{-1}$ mixture of two anti-GFP monoclonal antibodies (7.1 and 13.1; Roche), or 500 ng ml$^{-1}$ anti-tetracycline repressor protein monoclonal antibody (9G9; Clontech), followed by 80 ng ml$^{-1}$ goat anti-mouse immunoglobulins conjugated to horseradish peroxidase (Sigma) in TBS-T containing 1% (w/v) skimmed milk. Antibodies were detected using Western Lightning enhanced chemiluminescence reagent (GE healthcare) captured on a Fusion FX Spectra (Vilber Lourmat).

## Flow cytometry

For quantitative analysis of fluorescence by flow cytometry, ~$3\times10^6$ cells were harvested by centrifugation at 1200 *g*, 5 min and resuspended in 120 μl PBS containing 1% (w/v) formaldehyde, 0.2% (w/v) glutaraldehyde and 10 μg ml$^{-1}$ propidium iodide. This suspension was incubated at room temperature for 10 minutes, after which time the suspension was diluted 5-fold with PBS and analysed on a FC500 flow cytometer (Beckman Coulter), collecting a >20,000 gated events for each sample. Events were gated on the basis of forward- and side-scatter to remove clumps of cells and debris, and only cells with intact plasma membrane (negative for propidium) were included for analysis.

## Testing of double-strand break induction

To test for loss of *HYG* due to induction of I-SceI endonuclease expression, cells were removed from hygromycin selection and cultured with or without 1 μg ml$^{-1}$ tetracycline for 48h. These cultures were then diluted in fresh culture medium without antibiotic, and distributed across 96 well plates at an estimated 1.5 cells well$^{-1}$. After growth for 7 days, plates were replicated by 5-fold dilution into plates containing fresh medium with or without 2 μg ml$^{-1}$ hygromycin B and cultured for a further 1.5 days. The number of wells that had become hygromycin-sensitive was used to derive the proportion of *HYG*-negative cells based on Poisson-distributed filling of wells (λ estimated from populated wells on initial plating) and Binomially-distributed loss of *HYG*. Confidence intervals of estimates were derived from the distribution of quantiles from $10^6$ simulated platings with resampling λ (Normally-distributed around each estimate) and a range of probability of loss from 0–1.

## Genotypic analysis of modified cell lines

For testing of genotypes by PCR, ~$10^5$ cells from clonal lines or populations were harvested, washed in PBS and subjected to hot alkaline lysis in 50 mM NaOH, 2 mM EDTA for 10 min at 95˚C, followed by neutralisation by addition of Tris-HCl pH 6.8 to a final concentration of 75 mM. Small samples of these preparations were used as templates in multiplex PCR using primers for specific bait regions to screen transformants.

## Infections with transgenic *T. congolense* and bioluminescence measurements

In vitro luciferase activity was detected using the Luciferase Reporter Gene Assay, high sensitivity (Roche) and luminescence detected using a Glomax 96 Microplate Luminometer (Promega). For infections, female BALB/c strain animals were used except where indicated. Bloodstream-form *T. congolense* LUC2 cells were obtained from an infected donor mouse at the peak of parasitaemia, diluted in PBS supplemented with 20 mM glucose and used to inoculate recipient mice by intravenous injection at total doses of 100, 1000, or 10,000 parasites. D-luciferin was administered to animals at a dose of 200 mg kg$^{-1}$, by intraperitoneal injection 10 minutes before imaging. The mice were allowed 3 minutes of movement before being anaesthetized and placed in the imaging chamber where anaesthesia was maintained during acquisition. In vivo bioluminescence was captured on an IVIS Spectrum Imaging System (Perkin Elmer) and quantified using Living Image software (Xenogen).

## Supporting information

**S1 Fig. Localisation of proteins by modification of endogenous loci in bloodstream-form *T. congolense*.** Representative fields of view are shown for cells expressing *T. congolense* ESP10

C-terminally tagged with superfolder GFP (ESP10-sfG), or Nopp140 N-terminally tagged with mStrawberry (mSt-Nopp140). Native fluorescence from tagged proteins is shown, alongside counter-staining with 4′,6-diamidino-2-phenylindole (DAPI; cyan). ESP10-sfG cells have been additionally stained with AlexaFluor 594-conjugated tomato lectin (TL), to highlight the flagellar pocket and endosomal machinery.
(TIF)

**S2 Fig. Levels of transgenic TetR produced in modified cells.** Figure shows full view of immunoblot membrane excerpted in Fig 2D containing transferred whole cell lysates. Ponceau S staining is shown as a control for loading, plus two exposures of the membrane immunoblotted with an anti-TetR monoclonal antibody. 'wt' indicates unmodified parental cells.
(TIF)

**S3 Fig. Distribution of cellular fluorescence in cell lines expressing transgenic GFP from endogenous loci or ectopic inducible loci, and in the presence of RNAi against *GFP*.** A) Violin plots of green fluorescence of cells expressing GFP from the tubulin locus (*Δtubb*::*GFP*) or from inducible minichromosomal loci in 0 (non-induced) or 1 μg ml$^{-1}$ (induced) tetracycline (see Figs 3A and 4A). B) Violin plots of green fluorescence of cells expressing GFP from the tubulin locus (*Δtubb*::*GFP*) in which RNA-interference against *GFP* had been induced (see Fig 5A and 5B). Median and range of 5% and 95% quantiles are shown by dot and bars, respectively.
(TIF)

**S4 Fig. Stability of phenotypes associated with knockdown of clathrin heavy chain to continuous passage or freeze-thaw of cell lines.** Proportions of cells with clear morphological defect (see Fig 6) are shown for cells with no RNAi construct (parental) and two independent clones of δ*chc* cells following 0 or 48 h induction with tetracycline. Clones had previously been grown in culture for 1 week or 8 weeks following transfection and selection, or frozen after 1 week in culture, stored under liquid nitrogen and then brought back into growth (freeze-thaw). Bonferroni adjusted p-values from proportions test are shown above columns (n/s: not significant; *: $p < 0.05$; ***: $p < 0.001$).
(TIF)

**S5 Fig. Infection dynamics of *T. congolense* LUC2 cells in different albino mouse strains.** Male or female BALB/c and male B6 Albino (*C57BL/6N-Tyr$^{cWTSI}$*) were infected with 1000 *T. congolense* LUC2 cells at time 0 and dynamics monitored by whole-animal bioluminescence. Data from 15 independent infections using male BALB/c and B6 Albino animals, and 4 infections with female BALB/c animals are shown.
(TIF)

**S1 Dataset. Sequences and graphical maps for all new constructs used in this work.** Annotated sequences are provided in GenBank format and were used to generate graphical maps.
(ZIP)

**S1 Table. All primers used for amplification of *T. congolense* targeting sequences.**
(XLSX)

**S2 Table. Primers used in quantitative RT-PCR.**
(XLSX)

**S1 Supporting Information. Data supporting figures in main document and supplement.**
(XLSX)

## Acknowledgments

We are grateful to Mike Barrett (University of Glasgow) and Wendy Gibson (University of Bristol) for providing bloodstream-form and procyclic IL3000 cells, respectively. pSmOx and pLew100::NLS-ISceI-HA were gifts of Steve Kelly (University of Oxford) and Nina Papavasiliou (German Cancer Research Center, Heidelberg), respectively. pLew100::NLS-ISceI-HA was obtained via AddGene (#21299). We thank Tom Miller (University of Nottingham) for assistance with immunoblotting. RNA sequencing and library preparation were performed by Nadine Holmes in the Deep Seq facility, University of Nottingham.

## Author Contributions

**Conceptualization:** Catarina Gadelha, Bill Wickstead.

**Data curation:** Catarina Gadelha, Bill Wickstead.

**Formal analysis:** Georgina Awuah-Mensah, Jennifer McDonald, Pieter C. Steketee, Delphine Autheman, Sarah Whipple, Simon D'Archivio, Gavin J. Wright, Liam J. Morrison, Catarina Gadelha, Bill Wickstead.

**Funding acquisition:** Gavin J. Wright, Liam J. Morrison, Catarina Gadelha, Bill Wickstead.

**Investigation:** Georgina Awuah-Mensah, Jennifer McDonald, Pieter C. Steketee, Delphine Autheman, Sarah Whipple.

**Project administration:** Catarina Gadelha, Bill Wickstead.

**Resources:** Simon D'Archivio, Cordelia Brandt, Simon Clare, Katherine Harcourt.

**Supervision:** Gavin J. Wright, Liam J. Morrison, Catarina Gadelha, Bill Wickstead.

**Validation:** Georgina Awuah-Mensah, Jennifer McDonald, Pieter C. Steketee, Sarah Whipple.

**Visualization:** Georgina Awuah-Mensah, Pieter C. Steketee, Delphine Autheman, Catarina Gadelha, Bill Wickstead.

**Writing – original draft:** Catarina Gadelha, Bill Wickstead.

**Writing – review & editing:** Georgina Awuah-Mensah, Jennifer McDonald, Pieter C. Steketee, Delphine Autheman, Sarah Whipple, Simon D'Archivio, Cordelia Brandt, Simon Clare, Katherine Harcourt, Gavin J. Wright, Liam J. Morrison, Catarina Gadelha, Bill Wickstead.

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
