## [Decision Letter · Decision Letter 0]

4 Nov 2020

Dear Dr. Gadelha,

Thank you very much for submitting your manuscript "Reliable, scalable functional genetics in bloodstream-form Trypanosoma congolense in vitro and in vivo" for consideration at PLOS Pathogens. As with all papers reviewed by the journal, your manuscript was reviewed by members of the editorial board and by several independent reviewers. The reviewers appreciated the attention to an important topic. Based on the reviews, we are likely to accept this manuscript for publication, providing that you modify the manuscript according to the review recommendations.

All three reviewers were positive about your paper, although the effectiveness of RNAi is somewhat disappointing. There are however some comments that warrant some extra attention:

a) How long were the siRNAs, and can you get a better knock-down with a longer one?

b) What is the expression level of the proteins of the RNAi pathway (or at least, levels of mRNA)? Different for different stages?

c) If you differentiate your RNAi lines to procyclic forms, how well (if at all) does the RNAi work? (You must have tried this...)

d) Have you tried transiently transfecting dsRNA?

Technical issue: Please state the number of replicates used to obtain error bars or confidence intervals. if it's small then the dot plots are much more informative.

Sincerely,

Christine Clayton

Associate Editor

PLOS Pathogens

David Sacks

Section Editor

PLOS Pathogens

Kasturi Haldar

Editor-in-Chief

PLOS Pathogens

orcid.org/0000-0001-5065-158X

Michael Malim

Editor-in-Chief

PLOS Pathogens

orcid.org/0000-0002-7699-2064

All three reviewers were positive about your paper, although the effectiveness of RNAi is somewhat disappointing. There are however some comments that warrant some extra attention:

a) How long were the siRNAs, and can you get a better knock-down with a longer one?

b) What is the expression level of the proteins of the RNAi pathway (or at least, levels of mRNA)? Different for different stages?

c) If you differentiate your RNAi lines to procyclic forms, how well (if at all) does the RNAi work? (You must have tried this...)

d) Have you tried transiently transfecting dsRNA?

Technical issue: Please state the number of replicates used to obtain error bars or confidence intervals. if it's small then the dot plots are much more informative.

Reviewer Comments (if any, and for reference):

Reviewer's Responses to Questions

**Part I - Summary**

Reviewer #1: The manuscript by Awuah-Mensah and co-workers describes the successful design and implementation of efficient genetic tools in the protozoan parasite Trypanosoma congolense. The authors build a system that allows for the first time the genetic manipulation of the bloodstream form of this parasite with high-efficiency. They also assembled a cell line that allows high-complexity library production that the research community can rapidly make use of. The manuscript is clearly written and it will be of value to the parasitology field. The tools they have developed will allow, among many others, to assess the functional properties of the many proteins whose functions are unknown; to dissect the pathways/genes associated with drug resistance/mode of action or virulence. In this sense, the creation of a bioluminescent cell line is a plus. In addition, the authors show that bloodstream form cells are somehow refractory to gene silencing via RNAi; however, they do not explore the possible causes nor inspect possible ways to overcome this issue. Still, I am supportive of publication, though make some suggestions for possible improvement of the manuscript.

Reviewer #2: Authors have described successful and better direct genetic modification of bloodstream-form T. congolense IL3000 which provides substantial contribution in trypanosomes research.

Authors describe a parental single marker line that expresses T. congolense-optimized T7 RNA polymerase and Tet repressor and show that minichromosome loci can be used as sites for stable, regulatable transgene expression with low background in non-induced cells. Using these tools, authors describe T. congolense-specific constructs for inducible RNAi and demonstrate knockdown of some essential and nonessential genes. They show that a minichromosomal site can be use to create a stable bloodstream-form line that robustly provides >40,000 independent stable clones per transfection – enabling the production of high complexity libraries of genome-scale. They also show that modified forms of T. congolense are still infectious, create stable high-bioluminescence lines that can be used in models of AAT, and follow the course of infections in mice by in vivo imaging.

The data outlined here focused on bloodstream-form cells, and although the mRNA and proteins efficiency is ~30-50% of parental levels, the tools and methods developed here can be adapted to other replicative stages. The data presented is of high importance to researchers in the trypanosomes field and it opens up the way for gene manipulations in the parasite.

Reviewer #3: In this paper, the authors describe a series of genetic tools that will considerably help the community to study Trypanosoma congolense, a parasite of significant veterinary impact. They have shown that transfection efficiency can be very high when a I-SceI break system is used (40.000 clones per transfection), they generated a single-marker cell-line TcoSM that allows very tight inducible expression of transgenes and created a construct to perform RNAi (p3T7-TcoV).

The manuscript is very clearly written. The experiments are well controlled and interpretations are clear and rigorous.

**Part II – Major Issues: Key Experiments Required for Acceptance**

Reviewer #1: None noted

Reviewer #2: none

Reviewer #3: The reduced efficiency of RNAi (to 30-50% of original levels) is somewhat frustrating, given that in T. brucei these values are normally lower (better). Perhaps the AGO/DICER machinery is not as effective in T. congolense as in T. brucei. Does analysis of transcriptomes of these two species reveal lower transcript levels of RNAi components in T. congolense? The authors could also transfect a dsRNA for tubulin and clathrin heavy chain (transiently) and test the efficiency of depletion. If it is modest as as with the stable genetic system, it would further suggest that RNAi may simply be less efficient in this parasite species. Although it sounds as a “negative result” that RNAi may not be a fantastic tool in T. congolense for non-essential genes, this is an important conclusion, which needs to be published.

As the authors mentioned in the Discussion, conditional knock-out might be a better strategy in T. congolense for loss of function studies. It would be important to show that conditional knock-out of tubulin or clathrin heavy chain works very efficiently. This would give the community a good solution for depleting genes of interest.

**Part III – Minor Issues: Editorial and Data Presentation Modifications**

Reviewer #1: 1 I found the legends all over the manuscript to be too brief; more details would be beneficial. What ‘TL’ means in Fig 1B? The manuscript is clearly written by and for people working on trypanosomes, so I consider that adding more details will potentially engage a wider reader audience.

2. The transfection protocol described is based on the Amaxa nucleofector which has been a real game changer in the field; however, not every lab has one. It would be beneficial if the author can compare efficiencies when standard electroporators (BTX, Biorad) are used. If too low, it may save time and money to researchers with low budgets willing to work on T. congolense bloodstream forms.

3. In the same line, the authors claim that its protocol is also useful to transfect PF; however, only one transfection is shown. I understand PF transfection has never been an issue but probably more examples would be clarifying (or rephrase it). In T. brucei for instance, the transfection of PF using Amaxa nucleofector doesn’t seem to be more efficient than the old & standard electroporator devices.

4. I consider Fig. 2 to be supplemental. As I see it, in every cell line both T7 and tetR are in vastly excess. While tetR dimers must bind to one or two sites, T7pol is highly processive. Independently of this, what is relevant and it is shown later in Fig. 3 is that the assembled cell lines are able to express/repress the transgene in response to the inducer presence/absence and that the fold-change is large enough.

5. Fig3A. Is the plasmid missing a promoter for the drug resistance or background (uninduced) transcription by T7 is enough for selection? If the latter is true, transcription may not be enough for selection with all drugs; clarify it. In Fig. 3B, it shows the VSG copy number on minichromosomes is “1” but then the authors realize that unfortunately is not the case; please correct the actual number (at least 3/uncertain). Fig. 3D shows the signal range obtained tagging GFP at different loci when induced with 1ug/ml tetracycline. Add a curve showing how the system behaves to different inducer concentrations. Although the chosen representation is informative by showing the total spectrum of values, it doesn't show the proportions of cells displaying such intensities; thus, a standard histogram/violin plot will be much more informative (as in the Suppl). In this sense, I would also prefer Fig. 1 (in situ tagging) to show an histogram/violin plot and not a pic with a single cell (add also a pic where more cells can be seen). Taken together I think not only the fold change should be considered to select the best locus as heterogeneity/proportion in the signal also affects the decision. This is especially true when the RNAi of diverse genes display such a heterogenous effect on the cell population. In fact, tagging at the DBRH (R) seems to be at the population level the best option (high signal and concentrated); however, the clones then showed larger distributions, that’s odd. Any idea?

6. Regarding the RNAi refractory behaviour of bloodstream forms, there is still plenty room for research. The authors have not checked (or shown) to what extent dsRNA or siRNA are actually produced. It is also known that increasing the dsRNA length (among other factors, mostly thermodynamics), increases the RNAi efficacy; however, it is not indicated how long are the dsRNA utilised here. I suppose the RNAi machinery components and functions are conserved between T. congolense/brucei; please comment on this.

7. In Fig. 7 the authors calculate the transfection efficiency using a single plasmid. The use of a library where plasmids will have different sizes, GC content and so on may affect this number; please comment on this. Fig 7C; I understand the legend is missing the integration of the tub RNAi construct. I had to go through these lines several times to understand what you did, please rephrase it.

8. Please comment about the quality of the reference genome IL3000. A well curated genome is vital to perform molecular biology. In this sense, have the authors tried to modify others T. congolense strains? Are they expected to be easily transfected?

9. How stable is the TcoTTS cell line? for how long have the authors grown/transfecting them without affecting efficiency? Such a stable cell line will save quite a lot of effort to the research community and should be an inspiration for the T. brucei one.

Reviewer #2: 1. Introduction

The introduction/ literature is very good except that the information that "T. congolense LIKELY causes the majority of disease in sub-Saharan cattle (2,3), as well as making up a substantial proportion of infections (4)" cannot be entirely true as the predominance of a particular species may vary from place to place. The use of ”MAY” should be more appropriate.

2. Results

a. In Figure 2D - the wells as seen in the PonceauS does not properly correspond to the bands seen on the western blot. It looks like some parts were cut off (WT) and the TcoSM band looks stretched out as if the sample was loaded in a thinner well.

b. In Figure 3C – Authors have indicated that “All lines were captured and processed equally, except that GFP signal however at a closer look it seems some of the cells looks relatively smaller than the others e.g. in DBRH (F) and (R) and 369bp repeat. why is this so? It will be useful to clarify the differences here.

c. Figure 4 – it will be useful to represent the % knockdown (quantification) below the upper gel 4D. It was not clear the criteria for choosing 48h induction in some cases and 72h inductions in other situations as even a longer RNAi had a relatively lower reduction compared to T. brucei (For example, in Figure 4D and E).

d. Also the RNAi against a further 5 endogenous genes (PPDK, TcIL3000.A.H_000922100; CHC, TcIL3000.A.H_000768200; FBPase, TcIL3000.A.H_000671500; FH1, TcIL3000.A.H_000909500; PEPCK, TcIL3000.A.H_000300300) consistently reduced mRNA levels, but only to 30-60% of parental or non-induced levels (Fig. 4E,F). However, there is increase in mRNA levels after 48hours for all transcripts 4F. Authors should clarify this in the time course experiment.

e. Figure 5 – is there a special reason/s for having the two different pictures at 48h lower and upper pictures? Do they give the same information? 5A. What are the error bars on 5C? how many independent experiments per cell line?

Methods

In writing references two systems are written. While most of the references are given by numbers, Line 435 has - TcPCF3 (14). Bloodstream-form T. brucei SMB cells (Wirtz et al., 1999) were grown in HMI-9 medium supplemented with 15% fetal bovine serum at 37ÅãC and 5% CO2 (43).

References

While some of the references have the month and day this is omitted in other (2, 5, 9, 12 etc).

Reviewer #3: To test the virulence of the transgenic T. congolense lines, authors infected BALB/c mice. Could the authors explain why they chose this strain of mice, instead of C57BL/6 mice?

PLOS authors have the option to publish the peer review history of their article (what does this mean?). If published, this will include your full peer review and any attached files.

Reviewer #1: **Yes: **Esteban D. Erben

Reviewer #2: No

Reviewer #3: No
---

## [Editor Report · Decision Letter 1]

7 Dec 2020

Dear Dr. Gadelha,

We are pleased to inform you that your manuscript 'Reliable, scalable functional genetics in bloodstream-form Trypanosoma congolense in vitro and in vivo' has been provisionally accepted for publication in PLOS Pathogens.

Best regards,

Christine Clayton

Associate Editor

PLOS Pathogens

David Sacks

Section Editor

PLOS Pathogens

Kasturi Haldar

Editor-in-Chief

PLOS Pathogens

orcid.org/0000-0001-5065-158X

Michael Malim

Editor-in-Chief

PLOS Pathogens

orcid.org/0000-0002-7699-2064

Thank you for modifying the manuscript, I hope you found the suggestions useful.
---

## [Editor Report · Acceptance letter]

17 Jan 2021

Dear Dr. Gadelha,

We are delighted to inform you that your manuscript, "Reliable, scalable functional genetics in bloodstream-form Trypanosoma congolense in vitro and in vivo," has been formally accepted for publication in PLOS Pathogens.

Best regards,

Kasturi Haldar

Editor-in-Chief

PLOS Pathogens

orcid.org/0000-0001-5065-158X

Michael Malim

Editor-in-Chief

PLOS Pathogens

orcid.org/0000-0002-7699-2064